# A Kernel-Based View of Language Model Fine-Tuning

## Abstract

It has become standard to solve NLP tasks by fine-tuning pre-trained language models (LMs), especially in low-data settings. There is minimal theoretical understanding of empirical success, e.g., why fine-tuning a model with $10^8$ or more parameters on a couple dozen training points does not result in overfitting. We investigate whether the Neural Tangent Kernel (NTK)—which originated as a model to study the gradient descent dynamics of infinitely wide networks with suitable random initialization—*describes* fine-tuning of pre-trained LMs. This study was inspired by the decent performance of NTK for computer vision tasks (Wei et al., 2022). We also extend the NTK formalism to fine-tuning with Adam. We present extensive experiments that show that once the downstream task is formulated as a language modeling problem through prompting, the NTK lens can often reasonably describe the model updates during fine-tuning with both SGD and Adam. This kernel view also suggests an explanation for success of parameter-efficient subspace-based fine-tuning methods. Finally, we suggest a path toward a formal explanation for our findings via Tensor Programs (Yang, 2020b).

## 1 Introduction

It is now customary to solve most supervised natural language processing (NLP) tasks such as topic classification and textual entailment by fine-tuning a pre-trained language model (e.g., (Devlin et al., 2019; Liu et al., 2020b; Clark et al., 2020; Raffel et al., 2020; Joshi et al., 2020)). We lack theoretical understanding of this fine-tuning paradigm. Why do we not see over-fitting when fine-tuning a very large language model using a couple dozen instances of the supervised task? Why is fine-tuning so sensitive to details such as whether or not we include a prompt (e.g., adding "It was [great/terrible]" for sentiment analysis (Schick & Schütze, 2021; Gao et al., 2021)? Why does restricting optimization to a low-rank subspace of model parameters (Hu et al., 2021; Li et al., 2018; Aghajanyan et al., 2021) still result in performance comparable to full fine-tuning? Answering such questions requires understanding how the sequence of parameter updates changes in various scenarios, e.g., the addition of a prompt, or the introduction of randomly initialized parameters. The current theory of deep learning, at first sight, seems too primitive to address such questions, especially since fine-tuning has to start from a parameter initialization inherited from pre-training.

Recently, Wei et al. (2022) suggested replacing fine-tuning with Neural Tangent Kernel (NTK), an idea invented for study of infinite-width deep neural networks (Jacot et al., 2018; Du et al., 2019a) and previously applied to solving vision tasks with infinitely wide ConvNets (Arora et al. (2019b)). They note that NTK can be defined for any neural model $f$ and any initialization $\theta_0$ by representing an input $\xi$ by the gradient it induces $\nabla f(\xi; \theta_0)$, which yields a kernel matrix:

$$\mathcal{K}(\xi, \xi') = \langle \nabla f(\xi; \theta_0), \nabla f(\xi'; \theta_0) \rangle. \tag{1}$$

This kernel is well-defined for any parameter vector $\theta_0$. However, for an infinite-width network initialized with $\theta_0$ sampled from a suitably-scaled Gaussians, it can be shown that the kernel matrix is unchanged during gradient descent, which turns the classification task into a form of kernel regression with respect to this kernel (Jacot et al., 2018). In the fine-tuning setting, however, the initialization $\theta_0$ is inherited from the pre-trained network, and not sampled from the Gaussian distribution. Nevertheless, Wei et al. (2022) found that kernel regression using this "empirical NTK" (eNTK) defined with the inherited $\theta_0$ performs well, achieving classification accuracy within $6\%$

absolute of actual fine tuning on several image recognition tasks. In other words, their work hints that mathematical understanding of the fine-tuning phenomenon (e.g., its sample efficiency) could go via the theory of kernel classifiers.

The current paper furthers an empirical and theoretical understanding of the pre-training (PT) and fine-tuning (FT) paradigm for NLP tasks. Our contributions are:

1. **We formally extend the standard NTK theory developed for gradient descent to characterize kernel-based dynamics when training with Adam (Section 4).** We propose and rigorously prove the correctness of a new kernel formula relying on the sign of the gradient to describe early-stage training (e.g., fine-tuning) with Adam.

2. **We perform an extensive empirical analysis on 12 diverse NLP tasks to reveal when and to what extent fine-tuning exhibits kernel behavior (Section 5).** We find that using a prompt is crucial for the eNTK to achieve good performance, suggesting that prompting induces a well-characterized optimization benefit for fine-tuning. Further experiments reveal that the trajectory of prompt-based FT can often be *described* by kernel-based dynamics when the eNTK succeeds. The eNTK often achieves comparable performance to FT in the prompt-based setting but struggles to solve multi-class and entailment tasks.

3. **We straightforwardly apply the kernel view of FT dynamics to formally analyze the success fine-tuning methods that update in a low-rank subspace of model parameters (e.g., LoRA, Hu et al. (2021)).** These results in Section 6 highlight how a kernel-based understanding of FT can aid in the practical design and theoretical analysis of efficient variants.

4. **We formally extend infinite-width analysis to account for a pre-trained initialization and characterize conditions under which fine-tuning can exhibit kernel behavior.** Using insights into the importance of prompting, we formally prove the existence of a rigorous mechanism through which prompt-based FT of complex architectures (e.g., Transformers) can exhibit kernel behavior (Section 7). Analysis proceeds in the context of networks whose widths go to infinity (i.e., through the Tensor Programs framework), but unlike standard infinite-width NTK theory, it allows a non-random initialization (i.e., one that results from pre-training).

## 2   RELATED WORK

**Kernel view of training.**   The infinite-width limit is a well-studied theoretical model for deep network optimization. Jacot et al. (2018) introduced NTK to capture training a deep and infinitely wide neural network from a random initialization. Subsequent experiments showed that the kernels underperformed for standard tasks (Arora et al., 2019b) but performed well on small datasets (i.e., hundreds of examples) (Arora et al., 2020). Many works (Allen-Zhu et al., 2019a;b; Arora et al., 2019a; Du et al., 2019b;a; Li & Liang, 2018; Zou et al., 2018; Cao & Gu, 2019) have since applied this lens to understand the optimization and generalization behavior of deep networks. However, these analyses of optimization and generalization do not directly apply to the pre-training and fine-tuning framework because (1) the network trained during FT is inherited and non-random; and (2) LMs are often trained with Adam, and the NTK formula only describes training an infinitely wide network with SGD. In this work, we handle the issue of a non-random (i.e., pre-trained) initialization by assuming that the pre-training task is sufficiently related to the downstream task (Definition 7.3), and we derive new kernels to model early-stage training with Adam (Section 4).

**Theory of self-supervised learning and transfer learning.**   Existing theoretical works on transfer learning focus on linear probing and use the representation to provide guarantees on various tasks related to the original training data (Du et al., 2021; Tripuraneni et al., 2020; Wu et al., 2020). Additionally, Saunshi et al. (2021) studied autoregressive language models to rigorously characterized why prompting can improve zero-shot task performance, but their analysis precludes an investigation of FT. We focus on the masked language model pretraining objective, but it is worth noting that there are many works (Saunshi et al., 2019; Tosh et al., 2021a;b; Lee et al., 2021; Tsai et al., 2021) studying transfer when pre-training with a contrastive objective. However, experiments on language modeling (Abnar et al., 2021) and contrastive learning (Saunshi et al., 2022) recently demonstrated that properties of transfer between self-supervised pre-training and supervised FT cannot be fully captured by model-agnostic analyses that directly relate the pre-training and downstream task errors. Kernel theory provides a principled optimization- and architecture-aware framework to analyze FT.

**Optimization of transformers.** Several works (Zhang et al., 2020; Liu et al., 2020a; Li et al., 2022) have documented issues with optimizing Transformer-based architectures with SGD instead of Adam. To study the unique properties of optimizing transformers with Adam, we derive a new kernel formula (Theorem 4.3) to capture early-stage training with Adam. We show results with this kernel and FT with Adam and SGD in Table 1.

**Variants of fine-tuning methods.** A standard way of fine-tuning pre-trained LMs as introduced in Radford et al. (2018); Devlin et al. (2019) is to add a linear classifier on top of a PT encoder and update all the parameters together. Subsequent work (Schick & Schütze, 2021; Gao et al., 2021) formulated downstream tasks as a language modeling problem (i.e., prompt-based FT) and demonstrated empirical suuccess in low-data scenarios (see Liu et al. (2022) for a comprehensive survey). Another line of research studies parameter-efficient fine-tuning methods in which only a subset of model parameters are updated (Lester et al., 2021; Ben Zaken et al., 2022; Li & Liang, 2021) or the parameters updates are restricted to a low-dimensional subspace (Hu et al., 2021; Aghajanyan et al., 2021). We show that good eNTK performance arises only when studying prompt-based FT in Section 5 (Figure 1) and we later show in Section 6 that subspace-based fine-tuning methods such as LoRA (Hu et al., 2021) have a simple interpretation through the kernel.

# 3 PRELIMINARIES

## 3.1 KERNEL BEHAVIOR

It has been mathematically proven that training infinitely wide deep networks (with large Gaussian initialization) on small datasets can cause deep learning to turn into kernel-based learning. We are interested in identifying kernel behavior arising when training from an arbitrary initialization. Below, we adapt the definition of *lazy regime* (Woodworth et al., 2020) to an arbitrary initialization.

**Definition 3.1** (Kernel Behavior). Consider a neural network $f(\xi; \theta)$ that takes input $\xi$ and computes a scalar output[1] using $\theta$ as the parameters. Let $\theta_t$ be the parameters after $t$ steps of training by a gradient-based optimization algorithm. We say this training process of the network demonstrates *kernel behavior* if the following properties are satisfied.

1. *Linearization*: The change of the network can be approximated by its first order Taylor expansion, i.e.,
$$f(\xi; \theta_t) - f(\xi; \theta_{t-1}) \approx \langle \nabla f(\xi; \theta_{t-1}), \theta_t - \theta_{t-1} \rangle;$$

2. *Fixed Features*: The gradient at step $t$ is approximately the same as before training, i.e.,
$$\nabla f(\xi; \theta_t) \approx \nabla f(\xi; \theta_0).$$

$\nabla f$ denotes the gradient of $f$ w.r.t. $\theta$. "Closeness to kernel behavior" is quantified using the difference in the quantities on the two sides of the $\approx$ symbol.

**Definition 3.2** (Kernel Analog). Suppose optimization of the parameters $\theta$ of a model $f$ using the gradient-based update algorithm $\mathcal{A}$ to minimize a loss $\ell : \mathbb{R}^2 \to \mathbb{R}$ exhibits kernel behavior (Definition 3.1). Then, we say that a kernel $\mathcal{K}^{(\mathcal{A})}$ is the *kernel analog* of the optimization algorithm $\mathcal{A}$ if
$$f(\xi; \theta_t) - f(\xi; \theta_{t-1}) \approx -\eta \chi(\xi_t, \theta_{t-1}) \mathcal{K}^{(\mathcal{A})}(\xi, \xi_t), \forall t \geq 0 \tag{2}$$
where $\xi_t$ is the training input of step $t$, $\theta_t$ is the parameter at step $t$, $\chi(\xi, \theta) = \frac{\partial \ell(f(\xi;\theta), y(\xi))}{\partial f}$ is the derivative of the loss with respect to the model output, and $y(\xi)$ is the label of $\xi$.

We illustrate the dynamics of an optimization algorithm that demonstrates kernel behavior relates to the kernel analog. Let $\mathcal{A}$ be stochastic gradient descent (SGD). If SGD exhibits kernel behavior, then we can write how the function changes for a chosen input $\xi$ as

$$f(\xi; \theta_{t+1}) - f(\xi; \theta_t) \approx \langle \nabla f(\xi; \theta_t), \theta_{t+1} - \theta_t \rangle = \langle \nabla f(\xi; \theta_t), -\eta \chi_t \nabla f(\xi_t; \theta_t) \rangle \approx -\eta \chi_t \mathcal{K}^{(\text{SGD})}(\xi, \xi_t),$$

---

[1] Note that for $C$-way classification, $f$ is a vector in $\mathbb{R}^C$. We say $f$ exhibits kernel behavior if the Linearization and Fixed Features properties hold for every component of $f$. The subsequent definition of a kernel analog also generalizes to a vector output component-wise.

where the approximations follow from the Linearization and Fixed Features property, respectively. This construction immediately yields the standard neural tangent kernel (NTK) formula for $\mathcal{K}^{(\text{SGD})}$ derived in Jacot et al. (2018), which represents an input $\xi$ as the resulting gradient $\nabla f(\xi; \theta_0)$.

**Definition 3.3** (Neural Tangent Kernel $\mathcal{K}^{(\text{SGD})}$). $\mathcal{K}^{(\text{SGD})}(\xi, \xi') = \langle \nabla f(\xi; \theta_0), \nabla f(\xi'; \theta_0) \rangle$

Given an kernel $\mathcal{K}$, one can solve the classification problem by learning kernel coefficients $\alpha_i$ to minimize the empirical risk of $\sum_i \alpha_i \mathcal{K}(\cdot, \xi_i)$, where $\{\xi_i\}$ is the training data (see Appendix A). In Section 4, we derive the kernel analog for SignGD (i.e., an early-stage approximation of Adam), and in Section 5, we compare its eNTK performance against Adam FT. The eNTK computation relies on two design choices for the setting: (1) what the model output $f(\xi; \theta)$ is, and (2) which optimization algorithm $\mathcal{A}$ is being studied. For a given setting, the eNTK can be computed directly using the kernel analog (Definition 3.2) of $\mathcal{A}$. We run experiments choosing $\mathcal{A}$ as SGD or Adam and choosing $f$ based on the fine-tuning setting.

### 3.2 PRE-TRAINING AND FINE-TUNING PARADIGM

We focus our attention on masked language models (MLMs), such as BERT (Devlin et al., 2019) and RoBERTa (Liu et al., 2020b), which are trained to minimize the cross-entropy loss on independently predicting masked tokens (i.e., a $|\mathcal{V}|$-way classification task, where $\mathcal{V}$ is the vocabulary). Given a text input $s$ of length $T$ from the pre-training distribution $\mathcal{S}_{\text{PT}}$, replace a small percentage (e.g., 15%) of tokens with [MASK] tokens. This masked input is then fed into the representation function $h : \mathcal{S}_{\text{PT}} \to T \times \mathbb{R}^n$ (e.g., a Transformer encoder) to produce a low-dimensional contextual embedding for each position in the input. The contextual embeddings are independently multiplied by a classifier head (i.e., word embeddings) $\Phi \in \mathbb{R}^{n \times |\mathcal{V}|}$ to produce logits that will be used to compute the probability of a token filling each masked position.

Using a PT model to solve downstream tasks effectively has been a highly active area of research. We focus on fine-tuning (FT) methods, which adapt the pre-trained model to a new input distribution $\mathcal{S}_{\text{FT}}$ using additional training on the $C$-way downstream classification task.

1. **Standard FT** (Devlin et al., 2019; Liu et al., 2020b): To solve a $C$-way downstream classification task, initialize and learn[2] a new classifier head $\Gamma : \mathbb{R}^n \to \mathbb{R}^C$ on top of the contextual embedding of the [CLS], denoted $h_{\text{[CLS]}}$. In this case, the choice of $f : \mathcal{S}_{\text{FT}} \to \mathbb{R}^C$ for the eNTK construction is $f(s) = \Gamma(h_{\text{[CLS]}}(s))$.

2. **Prompt-based FT** (Schick & Schütze, 2021; Gao et al., 2021): Add a natural language prompt (e.g. "This is [MASK].") in addition to the task input to the downstream task, and use the pre-trained MLM to fill in the masked token. Compute the logits over task-relevant words (e.g., "great" and "terrible") using the corresponding columns of $\Phi$, denoted $\tilde{\Phi} \in \mathbb{R}^{n \times C}$. These logits will serve as surrogates to solve the downstream task. In this case, the choice of $f : \mathcal{S}_{\text{FT}} \to \mathbb{R}^C$ for the eNTK construction is $f(s) = \Phi^\top h_{\text{[MASK]}}(s)$.

## 4 KERNEL DERIVATION FOR ADAM

Computing the eNTK requires using the kernel analog (Definition 3.2) of the chosen optimization algorithm $\mathcal{A}$. However, it is difficult to construct a long-term kernel analog for Adam, because the adaptivity causes the each update to depend on the entire gradient history. Previous work has shown that in the early stages of training, full-batch (Ma et al., 2022) and mini-batch (Malladi et al., 2022) Adam updates can be approximated as using the sign of the gradient. In particular, the moving averages for the moment estimates are computed in a small neighborhood when the learning rate is small, so the Adam update is similar to performing coordinate-wise normalization on the gradient. This gradient-based optimization algorithm is called SignGD, defined below.

**Definition 4.1** (SignGD). SignGD is a gradient-based optimization algorithm that updates the parameters as $\theta_{t+1} = \theta_t - \eta \operatorname{sign}(\nabla f(\xi_t; \theta_t))$, where sign is applied element-wise.

---

[2]In our experiments, Standard FT corresponds to initializing $\Gamma$ at the linear probing solution (i.e., training $\Gamma$ on the downstream task while freezing all other layers) and then performing FT. We do this because when FT exhibits kernel behavior (Definition 3.1), it finds solutions close to initialization, and we hypothesize that the $\Gamma$ learned during FT is closer to the linear probing solution than a random initialization.

We define the sign-based kernel below and prove that it is the correct kernel analog for SignGD.

**Definition 4.2** (Asymmetric SignGD Kernel). $\mathcal{K}^{\text{(A-SignGD)}}(\xi, \xi') = \langle \nabla f(\xi; \theta_0), \text{sign}(\nabla f(\xi'; \theta_0)) \rangle$.

**Theorem 4.3** (Informal version of Theorem C.4). *If a network is trained with SignGD (Definition 4.1) and the training exhibits kernel behavior (Definition 3.1), then the training dynamics can be characterized as*

$$f(\xi; \theta_t) - f(\xi; \theta_{t-1}) \approx -\eta \chi_t \mathcal{K}^{\text{(A-SignGD)}}(\xi, \xi_t),$$

*where $\chi_t$ is the derivative of the loss with respect to $f$ at step $t$.*

*Proof sketch.* By the Linearization property in Definition 3.1,

$$f(\xi; \theta_t) - f(\xi; \theta_{t-1}) \approx \langle \nabla f(\xi; \theta_t), \theta_t - \theta_{t-1} \rangle = -\eta \chi_t \langle \nabla f(\xi; \theta_t), \text{sign}(\nabla f(\xi_t; \theta_{t-1})) \rangle.$$

Then by the Fixed Features in Definition 3.1,

$$\langle \nabla f(\xi; \theta_t), \text{sign}(\nabla f(\xi_t; \theta_{t-1})) \rangle \approx \langle \nabla f(\xi; \theta_0), \text{sign}(\nabla f(\xi_t; \theta_0)) \rangle = \mathcal{K}^{\text{(A-SignGD)}}(\xi, \xi_t). \qquad \square$$

We solve the asymmetric kernel regression by building an augmented system modified from He et al. (2022b) (Appendix A.3), but the difficulties of solving the kernel regression problem with an asymmetric kernel motivate us to also use the symmetric SignGD kernel, though it is not as theoretically sound as the asymmetric one.

**Definition 4.4** (SignGD Kernel). $\mathcal{K}^{\text{(SignGD)}}(\xi, \xi') = \langle \text{sign}(\nabla f(\xi; \theta_0)), \text{sign}(\nabla f(\xi'; \theta_0)) \rangle$

The kernel analog for Adam differs from the standard NTK formula for SGD because the sign function is agnostic to the relative scales of the gradients.

## 5 EXPERIMENTS

We compute the eNTK as described in Section 3 for different optimization algorithms and FT settings. eNTK performance being comparable to FT performance is a necessary but not sufficient condition for FT to exhibit kernel behavior (Definition 3.1), so we also directly measure if the Linearization and Fixed Features properties hold (Section 5.3). Overall, we find that only prompt-based FT exhibits kernel behavior, although the eNTK still struggles with multi-class classification and exhibits anomalous behavior on entailment tasks.

### 5.1 SETUP

Our experiments follow the few-shot setting from Gao et al. (2021) and use their manual prompt templates. We consider 12 NLP tasks, divided equally into 6 single sentence and 6 sentence pair datasets, which cover: sentiment analysis (SST-2, MR, CR); classifying an opinion's polarity (MQPA) or subjectivity (Subj) or question type (TREC); natural language inference (MNLI, SNLI, QNLI, RTE); and paraphrase detection tasks (MRPC, QQP). For each task, we randomly sample 5 $k$-shot datasets with $k$ training examples for each label. We use a pre-trained RoBERTa-base (Liu et al., 2020b) for FT and eNTK. Appendix A contains further details on datasets and implementation.

### 5.2 KERNEL PERFORMANCE ON DOWNSTREAM TASKS

**Prompting is critical for eNTK to match FT performance.** We measure the eNTK performance in the standard and prompt-based FT settings across SST-2, MR, QNLI, QQP (Figure 1). In the standard FT setting, $\mathcal{K}^{\text{(SGD)}}$ and SGD FT demonstrate a gap of up to $15\%$ absolute on tasks that exhibit only a $1\%$ gap in the prompt-based setting.

**SGD performs comparably to Adam in prompt-based FT.** We focus on the prompt-based FT setting (Table 1). We note that when doing prompt-based FT, Adam and SGD perform within 4% absolute of each other, furthering the discussion around the optimization of transformers with Adam versus SGD (Li et al., 2022; Zhang et al., 2020; Liu et al., 2020a).

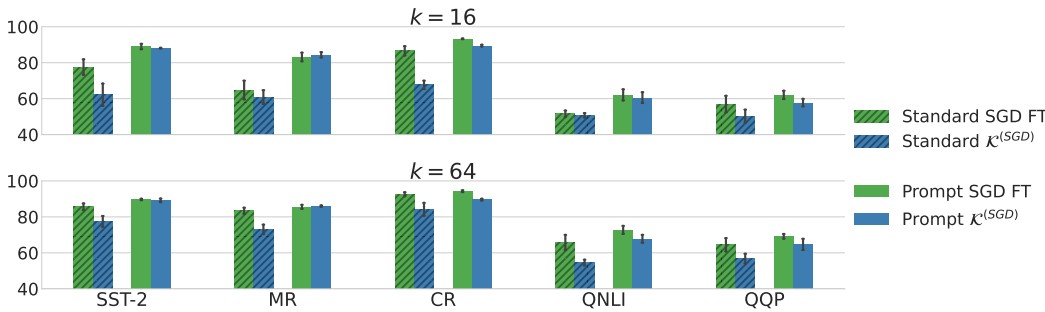

Figure 1: Comparing SGD-FT and $\mathcal{K}^{(\text{SGD})}$ performance in the standard and the prompt-based FT settings (Section 3) suggests that kernel behavior (Definition 3.1 can only arise when using a prompt.

**Prompt-based eNTK matches FT in most tasks.** To study if the eNTK can solve a given task in the prompt-based FT setting, we compare SGD-FT to $\mathcal{K}^{(\text{SGD})}$ and Adam-FT to $\mathcal{K}^{(\text{A-SignGD})}$ and $\mathcal{K}^{(\text{SignGD})}$ in Table 1. We observe that for 8 out of 12 tasks, $\mathcal{K}^{(\text{SGD})}$ can achieve accuracy within 5% absolute of SGD FT for $k = 16$ and $k = 64$. Similarly, $\mathcal{K}^{(\text{SignGD})}$ or $\mathcal{K}^{(\text{A-SignGD})}$ can achieve accuracy comparable to Adam FT for 7 out of the 12 tasks. The difference between $\mathcal{K}^{(\text{SignGD})}$ and $\mathcal{K}^{(\text{A-SignGD})}$ is negligible on most tasks. We suggest the asymmetry of the latter may cause $\mathcal{K}^{(\text{A-SignGD})}$ to sometimes perform worse than $\mathcal{K}^{(\text{SignGD})}$ despite being the theoretically sound kernel analog (Theorem 4.3).

**eNTK struggles with multi-class tasks.** The eNTK performs much worse than FT on all of the multi-class (i.e., TREC, MNLI, and SNLI) tasks, which we believe warrants further investigation. We conjecture that the eNTK cannot solve MNLI and SNLI despite solving other entailment tasks (i.e., QNLI and RTE) because the prompt is less natural when considering the label word "Maybe". One explanation of why the kernel analog sometimes outperforms FT is that certain batches may induce anomalous gradients that disrupt the FT trajectory, the effect of which the kernel can mitigate by downweighting these examples.

### 5.3 MEASURING KERNEL BEHAVIOR

The eNTK matches the performance of prompted FT for many tasks (Table 1), suggesting that these tasks may induce kernel behavior (Definition 3.1). However, the kernel's success may just be a coincidence. We take additional measurements to provide further empirical evidence that FT could be modeled as kernel behavior.

**Measuring the Linearization Property of Kernel Behavior** If FT exhibits kernel behavior (Definition 3.1), then the function output after FT should be close to the first order Taylor expansion around the pre-trained model:

$$f(\xi; \theta_{\text{FT}}) \approx f(\xi; \theta_{\text{PT}}) + \langle \nabla f(\xi; \theta_{\text{PT}}), \theta_{\text{FT}} - \theta_{\text{PT}} \rangle$$

where $\theta_{\text{PT}}$ is the model parameters after pre-training, $\theta_{\text{FT}}$ is the model parameters after fine-tuning on the downstream task, and $\xi$ is sampled from the test set. Figure 2 summarizes the results.

Pre-trained models perform significantly better than random on many single-sentence downstream tasks (e.g., SST-2, MR, and CR) but close to random on most sentence-pair tasks (e.g., QNLI, RTE, MRPC, and QQP). Subj, MNLI, and SNLI are outliers to this trend. The linearized model recovers a substantial amount of FT performance for SST-2, MR, CR, Subj, RTE, and QQP, all of which the eNTK could solve (Table 1).

Although pre-trained models perform much better than random on MNLI and SNLI, we find that the eNTK cannot solve these tasks very well (Table 1). Similarly, although the pre-trained model demonstrates near-random performance on QNLI and RTE, we find that the eNTK can solve these tasks. Moreover, although QNLI and RTE could be solved by the eNTK, the results suggest they do not induce the Linearization property of kernel behavior very strongly. Altogether, these findings suggest a deeper mystery around entailment tasks in particular.

**Measuring the Fixed Features Property of Kernel Behavior**    We also empirically test if the Fixed Features property (Definition 3.1) holds for tasks that the eNTK can solve. We measure the relative distance between $\mathcal{K}^{(\text{SGD})}$ computed before and after FT and record the average element-wise distance in Table 5. A smaller distance indicates that the Fixed Features property is more likely to hold. We see that tasks that the eNTK can solve exhibit relatively low (i.e., less than 1) distances.

| $k$-shot | Method | SST-2 | MR | CR | MPQA | Subj | TREC |
|---|---|---|---|---|---|---|---|
| 16 | SGD-FT | $\mathbf{89.0}_{(1.5)}$ | $83.2_{(2.4)}$ | $\mathbf{93.3}_{(0.2)}$ | $\mathbf{83.3}_{(1.3)}$ | $88.5_{(2.6)}$ | $\mathbf{80.3}_{(7.2)}$ |
| | $\mathcal{K}^{(\text{SGD})}$ | $88.3_{(0.3)}$ | $\mathbf{84.7}_{(1.5)}$ | $89.5_{(0.5)}$ | $76.4_{(2.7)}$ | $\mathbf{88.6}_{(1.3)}$ | $56.0_{(9.2)}$ |
| | Adam-FT | $\mathbf{88.3}_{(1.2)}$ | $81.3_{(6.1)}$ | $\mathbf{93.0}_{(1.6)}$ | $\mathbf{82.8}_{(2.2)}$ | $87.4_{(2.1)}$ | $\mathbf{79.6}_{(6.1)}$ |
| | $\mathcal{K}^{(\text{SignGD})}$ | $\mathbf{88.3}_{(0.5)}$ | $84.3_{(1.5)}$ | $89.0_{(4.0)}$ | $76.7_{(3.3)}$ | $\mathbf{89.2}_{(2.0)}$ | $58.9_{(7.1)}$ |
| | $\mathcal{K}^{(\text{A-SignGD})}$ | $\mathbf{88.3}_{(0.4)}$ | $\mathbf{84.9}_{(1.1)}$ | $88.0_{(1.8)}$ | $74.6_{(3.5)}$ | $88.6_{(1.8)}$ | $20.0_{(2.5)}$ |
| 64 | SGD-FT | $\mathbf{89.7}_{(0.4)}$ | $85.6_{(1.1)}$ | $\mathbf{94.3}_{(0.5)}$ | $84.8_{(0.8)}$ | $\mathbf{92.9}_{(0.5)}$ | $\mathbf{93.2}_{(1.0)}$ |
| | $\mathcal{K}^{(\text{SGD})}$ | $89.2_{(1.0)}$ | $\mathbf{86.4}_{(0.6)}$ | $89.8_{(0.3)}$ | $81.2_{(0.9)}$ | $91.4_{(0.7)}$ | $77.8_{(2.3)}$ |
| | Adam-FT | $\mathbf{89.3}_{(0.7)}$ | $\mathbf{86.0}_{(0.4)}$ | $\mathbf{93.7}_{(0.8)}$ | $\mathbf{84.6}_{(0.9)}$ | $\mathbf{92.7}_{(0.6)}$ | $\mathbf{92.6}_{(1.3)}$ |
| | $\mathcal{K}^{(\text{SignGD})}$ | $89.1_{(0.5)}$ | $85.6_{(1.0)}$ | $90.0_{(0.2)}$ | $78.6_{(6.4)}$ | $92.4_{(0.5)}$ | $82.0_{(1.4)}$ |
| | $\mathcal{K}^{(\text{A-SignGD})}$ | $88.9_{(0.9)}$ | $85.6_{(1.0)}$ | $90.1_{(0.7)}$ | $81.8_{(1.1)}$ | $91.8_{(1.1)}$ | $21.0_{(4.3)}$ |

(a) Single-sentence tasks

| $k$-shot | Method | MNLI | SNLI | QNLI | RTE | MRPC | QQP |
|---|---|---|---|---|---|---|---|
| 16 | SGD-FT | $\mathbf{59.2}_{(2.7)}$ | $\mathbf{65.7}_{(2.7)}$ | $\mathbf{62.1}_{(3.1)}$ | $\mathbf{60.0}_{(5.5)}$ | $\mathbf{73.9}_{(2.7)}$ | $\mathbf{62.1}_{(2.3)}$ |
| | $\mathcal{K}^{(\text{SGD})}$ | $53.0_{(3.0)}$ | $57.8_{(2.3)}$ | $60.1_{(3.3)}$ | $\mathbf{60.0}_{(4.7)}$ | $73.4_{(5.6)}$ | $58.2_{(0.9)}$ |
| | Adam-FT | $\mathbf{56.8}_{(2.9)}$ | $\mathbf{64.6}_{(4.1)}$ | $\mathbf{63.1}_{(3.5)}$ | $57.6_{(6.3)}$ | $\mathbf{77.6}_{(3.1)}$ | $\mathbf{61.8}_{(4.5)}$ |
| | $\mathcal{K}^{(\text{SignGD})}$ | $53.8_{(1.2)}$ | $53.7_{(1.5)}$ | $59.5_{(3.1)}$ | $55.4_{(4.2)}$ | $75.6_{(1.2)}$ | $60.7_{(2.2)}$ |
| | $\mathcal{K}^{(\text{A-SignGD})}$ | $51.9_{(4.0)}$ | $54.9_{(3.1)}$ | $56.0_{(1.9)}$ | $\mathbf{59.8}_{(4.0)}$ | $75.2_{(2.6)}$ | $59.4_{(2.0)}$ |
| 64 | SGD-FT | $\mathbf{68.7}_{(1.7)}$ | $\mathbf{77.3}_{(0.9)}$ | $\mathbf{72.8}_{(2.2)}$ | $\mathbf{68.9}_{(2.5)}$ | $\mathbf{82.8}_{(1.2)}$ | $69.2_{(1.3)}$ |
| | $\mathcal{K}^{(\text{SGD})}$ | $60.4_{(1.8)}$ | $65.5_{(1.6)}$ | $67.3_{(1.6)}$ | $66.5_{(2.5)}$ | $79.2_{(2.5)}$ | $66.4_{(1.7)}$ |
| | Adam-FT | $\mathbf{67.9}_{(1.0)}$ | $\mathbf{76.9}_{(1.4)}$ | $\mathbf{74.2}_{(3.2)}$ | $\mathbf{67.3}_{(2.7)}$ | $\mathbf{80.9}_{(1.2)}$ | $\mathbf{69.8}_{(0.6)}$ |
| | $\mathcal{K}^{(\text{SignGD})}$ | $59.9_{(1.7)}$ | $63.9_{(2.5)}$ | $65.4_{(1.7)}$ | $63.8_{(1.8)}$ | $77.4_{(2.3)}$ | $63.7_{(4.4)}$ |
| | $\mathcal{K}^{(\text{A-SignGD})}$ | $58.5_{(1.7)}$ | $66.8_{(1.1)}$ | $66.5_{(1.1)}$ | $63.8_{(2.2)}$ | $77.3_{(2.0)}$ | $66.1_{(3.4)}$ |

(b) Sentence-pair tasks

Table 1: Performance achieved by prompt-based FT and prompt-based eNTKs with different formulas on the LM-BFF test set (Gao et al., 2021). The eNTK performs comparably to the analogous FT on many tasks but fails on multi-class tasks (i.e., TREC, SNLI, and MNLI). Performance is measure by average test accuracy over 5 $k$-shot splits for all tasks except MRPC and QQP, where it is F1.

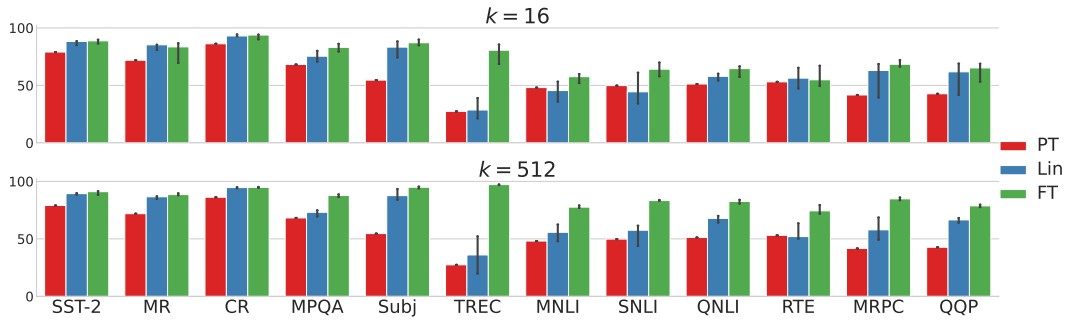

Figure 2: Accuracies of zero-shot pre-trained model (PT), linearized model (Lin., see Definition 3.1) and fine-tuned model (FT). Tasks that induce the Linearization property of kernel behavior (Definition 3.1) will show that Lin. performance recovers a substantial amount of the FT performance. For each $k$, we plot the median and range of the test accuracies across 5 seeds and data splits.

## 6 Efficacy of Subspace-Based Fine-Tuning Methods

We study parameter-efficient fine-tuning methods that roughly preserve performance but reduce the overhead of fine-tuning and saving a large language model (He et al., 2022a). One such method is LoRA (Hu et al., 2021), which restricts fine-tuning updates to be low-rank as defined below.

**Definition 6.1** ($\mathcal{A}$-LoRA FT (Hu et al., 2021)). Let $\mathcal{A}$ be a gradient-based optimization algorithm. For every weight matrix $W \in \mathbb{R}^{m \times n}$, choose $k \ll m$ and initialize $A \in \mathbb{R}^{m \times k}$ with i.i.d. mean-0 Gaussian values and $B \in \mathbb{R}^{k \times n}$ as 0. Set the weight to be $W + AB$. To fine-tune, fix $W$ at its pre-trained value and train only $A$ and $B$ using $\mathcal{A}$.

We also consider a fine-tuning variant that projects the parameter vector to a low-dimensional subspace. This method was originally proposed to characterize the difficulty of downstream tasks (Li et al., 2018), and recent LM experiments in Aghajanyan et al. (2021) have shown that fine-tuning the projected parameters can recover most of the performance of standard fine-tuning.

**Definition 6.2** ($\mathcal{A}$-IntrinsicDimension FT (Li et al., 2018; Aghajanyan et al., 2021)). Fix a random projection $\Pi \in \mathbb{R}^{M \times k}$, where $M$ is the number of parameters in a model $f$. To fine-tune using a loss $\ell$ on a downstream task, replace the gradient in the update formula of $\mathcal{A}$ with $\Pi^\top \nabla \ell(\xi; \theta)$.

Although theoretical characterization of these methods seems complex, the kernel view admits a simple interpretation. We straightforwardly apply the classical Johnson-Lindenstrauss, or JL, lemma in Johnson (1984), which guarantees inner product preservation under random projections, to show that these methods approximately preserve the SGD kernel (Definition 3.3).

**Theorem 6.3** (LoRA and IntrinsicDimension FT preserve $\mathcal{K}^{(SGD)}$, informal version of Theorem D.5). *Let $\mathcal{K}^{(SGD)}$ be the kernel analog (Definition 3.2 to SGD FT, $\mathcal{K}^{(SGD)}_{LoRA}$ be the kernel analog to SGD-LoRA FT (Definition 6.1), and $\mathcal{K}^{(SGD)}_{ID}$ be the kernel analog to SGD-IntrinsicDimension FT on a downstream task $\Xi$. Then, with high probabililty, $(\mathcal{K}^{(SGD)}_{LoRA}(i,j) - \mathcal{K}^{(SGD)}(i,j))/\mathcal{K}^{(SGD)}(i,j) \approx 0$ and $\mathcal{K}^{(SGD)}_{ID}(i,j) \approx \mathcal{K}^{(SGD)}(i,j)$ for all $i, j \in [N]$.*

*Proof sketch.* Consider an individual layer in the network and inputs $\xi, \xi' \in \Xi$ to the downstream task. LoRA randomly projects $\nabla_B f(\xi; \theta)$ and $\nabla_B f(\xi'; \theta)$, where $\nabla_B$ denotes the gradient with respect to $B$, and does not modify the gradient to $A$, since $B$ is initialized to zero. The rest of the proof for LoRA and the proof for IntrisicDimension FT follows from applying JL to all such pairs $\xi, \xi'$ to show the inner product, which determines the kernel entry, is preserved. $\square$

*Remark* 6.4. Theorem 6.3 states that the kernel analog of SGD FT is unchanged by LoRA in both prompt-based and standard FT. Therefore, the theorem only applies when $\mathcal{A}$ FT exhibits kernel behavior, which we find to only be in the prompt-based setting (Figure 1).

Experimental results in Table 7 verify that prompted SGD FT and prompted LoRA-SGD FT achieve similar performance on several downstream tasks, and $\mathcal{K}^{(SGD)}_{LoRA}$ achieves performance similar to $\mathcal{K}^{(SGD)}$. We leave it to future work to account for the success of these methods when FT does not exhibit kernel behavior.

## 7 Theory: Prompt-Based Fine-Tuning Can Exhibit Kernel Behavior

We give a plausible mechanism for how prompt-based FT can exhibit kernel behavior (Definition 3.1) as the network width grows large. We start by defining a pre-training scheme, which formalizes how changing the architecture width impacts pre-training.

**Definition 7.1** (Pre-Training Scheme). A pre-training scheme $(\mathcal{X}, \mathcal{A}, \mathcal{F}^n)$ with width $n$ contains the dataset $\mathcal{X}$, optimizer $\mathcal{A}$ and its hyperparameters, and a model architecture $\mathcal{F}^n$. Let $f^n \sim (\mathcal{X}, \mathcal{A}, \mathcal{F}^n)$ denote a model resulting from training the architecture $\mathcal{F}^n$ on the dataset $\mathcal{X}$ with optimizer $\mathcal{A}$.

*Remark* 7.2. The concrete reliance of the architecture on the width is given by Tensor Programs: for example, in a Transformer, increasing $n$ corresponds to increasing the embedding dimension.

We now connect pre-training to the downstream task. Analogous to Saunshi et al. (2021), we reason that prompting transforms the downstream task into a fill-in-the-blank problem, and thus the

downstream task can be viewed as a subcase of the pre-training task. We then assume that a wider pre-trained network will be better at filling in masked tokens and that an infinitely wide pre-trained network can solve the downstream task perfectly when using a suitable prompt.

**Definition 7.3** (Natural Task in the Infinite-Width Limit). We say that a downstream task $\Xi$ is natural with respect to a pre-training scheme $(\mathcal{X}, \mathcal{A}, \mathcal{F}^n)$ if, for any $f^n \sim (\mathcal{X}, \mathcal{A}, \mathcal{F}^n)$ and any $\xi \in \Xi$,

$$\lim_{n \to \infty} \chi(\xi, f^n(\xi)) = 0. \tag{3}$$

*Remark* 7.4. Note that a task may only be natural in the infinite-width limit when using a prompt, since standard FT will always require training a randomly initialized head (i.e., $\chi$ will not vanish at infinite width).

Although Definition 7.3 is asymptotic, we design a cheap empirical test. We require access to two models of different widths resulting otherwise identical pre-training schemes: $f^{n_1} \sim (\mathcal{X}, \mathcal{A}, \mathcal{F}^{n_1})$ and $f^{n_2} \sim (\mathcal{X}, \mathcal{A}, \mathcal{F}^{n_2})$. Then, we can check if $\chi$ decreases with width by measuring $\chi(\xi, f^{n_1}(\xi))$ and $\chi(\xi, f^{n_2}(\xi))$, with $n_1 \neq n_2$, without making any gradient updates (Table 8).

To study the behavior of fine-tuning one also needs to make assumptions about parameters that resulted from the pre-training. In particular, we assume that the network can be written as a Tensor Program (Yang, 2019; 2020a;b), which is sufficiently general to allow our theory to describe many complex architectures (e.g., Transformers). To allow the analysis to proceed by way of Tensor Programs, we require that the network is (1) *stable*: its output does not grow with width (i.e., the infinite-width limit is meaningful), and (2) *non-trivial*: its output can be updated during fine-tuning (i.e., learning can occur).

**Theorem 7.5** (Informal version of Theorem C.5). *Assume the downstream task $\Xi$ is natural in the infinite-width limit with a pre-trained model $f$, and $f$ is stable, non-trivial, and can be written as a Tensor Program. Then prompt-based FT of $f$ will exhibit the Linearization and Fixed Features properties of kernel behavior (Definition 3.1).*

The proof of the theorem formalizes the intuition that if the pre-trained network is already decent at solving the downstream task, the network needs to only mildly adapt to solve the downstream task. Notably, we extend standard NTK theory to account for an arbitrary initialization and to characterize early-stage training with Adam (see Section 4 for kernel).

## 8 CONCLUSION

We use NTKs to mathematically formalize the general intuition that fine-tuning pretrained language models to solve downstream tasks requires only a "small change." Extensive experiments on 12 NLU tasks demonstrate that prompt-based FT is much more likely to exhibit kernel behavior (Definition 3.1) than standard FT (Figure 1). Further experiments in the prompt-based FT setting using a newly derived kernel for Adam (Definition 4.2, see Theorem 4.3) demonstrate that the eNTK can match the performance of FT on many tasks. On the tasks that eNTK can solve, measurements in Section 5.3 suggest that prompt-based FT does exhibit kernel behavior. We demonstrate one possible use of the kernel view to explain empirical phenomena by applying it to understand subspace-based fine-tuning methods (Section 6), and we note that the kernel has many mathematically useful properties that can aid design and study of parameter-efficient fine-tuning methods. Moreover, one can use the kernel to study the inductive bias of FT, as was done for gradient descent from a random initialization in the past (Allen-Zhu et al., 2019b;a; Li & Liang, 2018). We provide a first-cut theoretical analysis in Section 7 as to why prompt-based fine-tuning can exhibit kernel behavior.

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

## A    EXPERIMENTAL DETAILS

### A.1    DATASETS AND PROMPTS

Table 2 shows the set of downstream tasks, which are adapted from Gao et al. (2021). We consider 6 single sentence classification datasets (SST-2 (Socher et al., 2013), MR (Pang & Lee, 2005), CR (Hu

| Dataset | $C$ | #Train | #Test | Type | Prompt | |
|---|---|---|---|---|---|---|
| SST-2 | 2 | 6,920 | 872 | sentiment | $<S_1>$ It was [MASK] . | {great, terrible} |
| MR | 2 | 8,662 | 1,000 | sentiment | $<S_1>$ It was [MASK] . | {great, terrible} |
| CR | 2 | 3,175 | 500 | sentiment | $<S_1>$ It was [MASK] . | {great, terrible} |
| MPQA | 2 | 8,606 | 1,000 | opinion polarity | $<S_1>$ It was [MASK] . | {great, terrible} |
| Subj | 2 | 8,000 | 1,000 | subjectivity | $<S_1>$ This is [MASK] . | {subjective, objective} |
| TREC | 6 | 5,452 | 500 | question cls. | [MASK] : $<S_1>$ | {Description, Expression, Entity, Human, Location, Number} |
| MNLI | 3 | 392,702 | 1,000 | NLI | $<S_1>$ ? [MASK] , $<S_2>$ | {Yes, Maybe, No} |
| SNLI | 3 | 549,367 | 1,000 | NLI | $<S_1>$ ? [MASK] , $<S_2>$ | {Yes, Maybe, No} |
| QNLI | 2 | 104,743 | 1,000 | NLI | $<S_1>$ ? [MASK] , $<S_2>$ | {Yes, No} |
| RTE | 2 | 2,490 | 277 | NLI | $<S_1>$ ? [MASK] , $<S_2>$ | {Yes, No} |
| MRPC | 2 | 3,668 | 408 | paraphrase | $<S_1>$ [MASK] , $<S_2>$ | {Yes, No} |
| QQP | 2 | 363,846 | 1,000 | paraphrase | $<S_1>$ [MASK] , $<S_2>$ | {Yes, No} |

Table 2: The statistics and prompts of the datasets we used in our experiments. The choices of prompts are from Gao et al. (2021) which include a template, and a set of label words that are used to fill in the [MASK] token. $<S_1>$ and $<S_2>$ refer to the first and the second (if any) input sentence.

& Liu, 2004), MPQA (Wiebe et al., 2005), Subj (Pang & Lee, 2004) and TREC (Voorhees & Tice, 2000)), and 6 sentence pair datasets (MNLI (Williams et al., 2018), SNLI (Bowman et al., 2015), QNLI (Rajpurkar et al., 2016), RTE (Dagan et al., 2005; Bar Haim et al., 2006; Giampiccolo et al., 2007; Bentivogli et al., 2009), MRPC (Dolan & Brockett, 2005) and QQP[3]. Our datasets represent 6/8 datasets of the GLUE benchmark Wang et al. (2019) (SST-2, MNLI, QNLI, RTE, MRPC, QQP).

In contrast to Gao et al. (2021), we make two modifications to the test sets. First, we split CR into 500 test examples and 3,175 training examples to ensure enough training examples for our 512-shot experiments and secondly, we limit the test sizes to 1,000 examples to speed up kernel evaluations.

To generate $k$-shot few-shot datasets, the original training data is used to randomly sample $K$ examples per label for training and another, separate $k$ examples per label for the validation set. Unless otherwise stated, we usually run experiments over 5 seeds of few-shot data sets. We directly use the 'manual' prompt templates and label words proposed by Gao et al. (2021), which are reproduced in Table 2.

## A.2 COMPUTING THE KERNEL

We adapt the functorch implementation of Novak et al. (2022) to compute the eNTK for a large model, using a mix of backward-mode auto-differentiation for computing the jacobians and forward-mode auto-differentiation for computing jacobian-vector products. Note that $\mathcal{K}^{(\text{SignGD})}$ cannot be computed using jacobian-vector products and thus requires significantly more memory and run-time in practice.

## A.3 SOLVING THE KERNEL

In the standard NTK setting, the initial output of the model $f(\cdot; \theta_0)$ contains no information about solving the task, because $\theta_0$ is a random initiaization. However, in the prompted FT setting, we expect the pre-trained model to be able to solve the downstream task well even before any fine-tuning occurs (see Table 4). So, we add the pre-trained model's output to the output from the kernel. Furthermore, we run a grid search over scaling the labels in order to take advantage of any pre-existing knowledge the model has about the downstream task. In particular, the kernel regression is based on the $\ell_2$ distance to the ground truth one-hot vector, but the pre-trained model outputs the logits which will be used for cross-entropy loss. Scaling the one-hot vector by $f_0$ helps align its scaling with the logits. Our hyperparameter grid for $f_0$ can be found in Table 3, where $\infty$ corresponds to not using the pre-trained model logits when solving the kernel.

**Solving Multi-Class Tasks** There are several options for how to solve $C$-way classification tasks ($C > 2$). We perform the most general one, which scales with $C^2$. Each logit is treated as an independent output of the network, essentially scaling the size $N$ of the original dataset by a factor of

---

[3] https://www.quora.com/q/quoradata/

$C$. With $CN$ examples, the kernel now has shape $CN \times CN$. The labels are also scaled up to treat the multi-class problem as many binary classification problems. Solving the multi-class task this way allows the kernel regression model to view relationships between different logits.

**Symmetric Kernel**  Given a symmetric kernel $\mathcal{K} \in \mathbb{R}^{N \times N}$, we solve the kernel regression problem. In particular, we use the representer theorem to write that the empirical risk minimizer of the loss can be expressed as a linear combination of the kernel features computed on the train set.

$$h^*(\cdot) = \arg\min_{h \in \mathcal{H}_\mathcal{K}} \frac{1}{N} \sum_{i=1}^N \ell(h(x_i), y_i) \quad \leftrightarrow \quad h^*(\cdot) = \sum_{i=1}^N \alpha_i \mathcal{K}(\cdot, x_i)$$

for a given loss function $\ell$. The symmetric SignGD and SGD kernels train $\alpha_i$ via gradient descent to minimize a regularized logistic loss on the downstream task. We search over a grid of regularization strengths chosen proportional to $\|\mathcal{K}\|_{\mathrm{op}}$, see Table 3. For a test input $x$, the kernel outputs the prediction $h(x) = \sum_i \alpha_i \mathcal{K}(x, x_i)$.

**Asymmetric Kernel**  We write how to solve the kernel regression problem with an asymmetric kernel, developed in He et al. (2022b), here. Consider the augmented linear system:

$$\begin{bmatrix} I/\gamma & H \\ H^\top & I/\gamma \end{bmatrix} \begin{bmatrix} \alpha \\ \beta \end{bmatrix} = \begin{bmatrix} 1 \\ 1 \end{bmatrix}$$

where $H_{ij} = y_i \phi_s(x_i)^\top \phi_t(x_j) y_j$ with $\phi_s$ and $\phi_t$ as the two different feature maps and $y_i$ as the label for the $i$th example. Define $\omega^*$ and $\nu^*$ as

$$\omega^* = \sum_i \beta_i^* y_i \phi_t(x_i)$$

$$\nu^* = \sum_i \alpha_i^* y_i \phi_s(x_i)$$

Solving this system yields two discriminant functions:

$$f_s(x) = K(x, X)(\beta^* \odot Y)$$
$$f_t(x) = K(X, x)(\alpha^* \odot Y)$$

where $K(x_i, x_j) = \langle \phi_s(x_i), \phi_t(x_j) \rangle$.

We can thus create one discriminant function as $c f_s(x) + (1 - c) f_t(x)$ where $c \in [0, 1]$ is some hyperparameter. When $\phi_s = \phi_t$, we see that $f_s = f_t$ and we reduce to the standard kernel problem (though with repeated equations). Note that per He et al. (2022b), this system is only meaningful in terms of stationary points when training $\alpha$ and $\beta$ using the least squares loss.

We now leverage some specific knowledge about the NTK setting. In particular, we know that we should only use $f_s$ as the predictor in order to correctly represent a new test input in the kernel analog for SignGD.

**Hyperparameters and Implementation**  We follow Gao et al. (2021) in using the few-shot validation set to search over hyperparameters and finding the best hyperparameter per few-shot dataset. We use value ranges given by Gao et al. (2021) and Hu et al. (2021), and search over a wider range of values for SGD. Table 3 shows the hyperparameter grids for fine-tuning and the kernel method.

Gao et al. (2021) train for 1000 steps in the 16-shot setting, and validate the performance every 100 steps to take the best checkpoints. As we consider varying values of $k$, we use the formula of training for $32kC$ steps and validating every $4kC$ steps, where $C$ is the number of classes in the dataset. This gives a comparable number of training and validation steps for binary tasks in the 16-shot setting.

## B  ADDITIONAL EXPERIMENTAL RESULTS

## C  KERNEL BEHAVIOR AND THE PARAMETRIZATION

As mentioned in Section 7, there are two behaviors of training neural networks — kernel behavior and feature learning behavior. Kernel behavior provides a tractable tool to study the training of

| Experiment | Hyperparameters | Values |
|---|---|---|
| SGD FT | Batch size | $\{2, 4, 8\} \times$ |
| | Learning rate | $\{1e-4, 5e-4, 1e-3, 5e-3, 1e-2\}$ |
| SGD-LoRA FT | Batch size | $\{4, 16\} \times$ |
| | Learning rate | $\{1e-4, 1e-3, 1e-2\} \times$ |
| | $(r_{LoRA}, \alpha_{LoRA})$ | $\{(8, 16)\}$ |
| Adam FT | Batch size | $\{2, 4, 8\} \times$ |
| | Learning rate | $\{1e-5, 2e-5, 5e-5\}$ |
| Adam-LoRA FT | Batch size | $\{4, 16\} \times$ |
| | Learning rate | $\{1e-5, 4e-5, 4e-4\} \times$ |
| | $(r_{LoRA}, \alpha_{LoRA})$ | $\{(8, 16)\}$ |
| $\mathcal{K}^{(SGD)}, \mathcal{K}^{(SignGD)}$ | Kernel regularization | $\{0, 0.001, 0.01, 0.1, 1\} \times$ |
| | $f_0$ scaling | $\{10, 100, 1000, 10000, \infty\}$ |
| $\mathcal{K}^{(A\text{-}SignGD)}$ | Kernel regularization | $\{0, 0.001, 0.01, 0.1, 1\} \times$ |
| | $f_0$ scaling | $\{10, 100, 1000, 10000, \infty\} \times$ |
| | Kernel $\gamma$ | $\{0.01, 0.1, 1, 10\} \times$ |
| | Kernel $c$ | $\{1\}$ |

Table 3: The hyperparameter grids used in our experiments.

| | SST-2 | | MR | | CR | |
|---|---|---|---|---|---|---|
| $k$-shot | Lin. | FT | Lin. | FT | Lin. | FT |
| 0 | —— 79.0 —— | | —— 71.9 —— | | —— 86.2 —— | |
| 16 | $87.5_{(1.3)}$ | $88.3_{(1.2)}$ | $84.3_{(1.8)}$ | $81.3_{(6.1)}$ | $93.3_{(0.6)}$ | $93.0_{(1.6)}$ |
| 64 | $88.6_{(0.4)}$ | $89.3_{(0.7)}$ | $85.0_{(0.2)}$ | $86.0_{(0.4)}$ | $94.0_{(0.5)}$ | $93.7_{(0.8)}$ |
| 512 | $89.2_{(0.5)}$ | $90.7_{(1.2)}$ | $86.3_{(0.8)}$ | $88.6_{(0.6)}$ | $94.0_{(0.5)}$ | $93.7_{(0.8)}$ |

| | MQPA | | Subj | | TREC | |
|---|---|---|---|---|---|---|
| $k$-shot | Lin. | FT | Lin. | FT | Lin. | FT |
| 0 | —— 68.2 —— | | —— 54.6 —— | | —— 27.4 —— | |
| 16 | $75.6_{(3.1)}$ | $82.8_{(2.2)}$ | $82.9_{(4.7)}$ | $87.4_{(2.1)}$ | $30.4_{(7.2)}$ | $79.6_{(6.1)}$ |
| 64 | $75.6_{(2.3)}$ | $85.0_{(0.2)}$ | $78.9_{(14.0)}$ | $92.7_{(0.6)}$ | $31.2_{(13.0)}$ | $92.6_{(1.3)}$ |
| 512 | $72.1_{(2.2)}$ | $87.7_{(0.8)}$ | $88.5_{(3.8)}$ | $94.8_{(0.5)}$ | $35.4_{(12.5)}$ | $97.1_{(0.3)}$ |

| | MNLI | | SNLI | | QNLI | |
|---|---|---|---|---|---|---|
| $k$-shot | Lin. | FT | Lin. | FT | Lin. | FT |
| 0 | —— 48.1 —— | | —— 49.8 —— | | —— 51.2 —— | |
| 16 | $43.6_{(6.4)}$ | $56.8_{(2.9)}$ | $47.2_{(9.3)}$ | $64.6_{(4.1)}$ | $57.5_{(2.3)}$ | $63.1_{(3.5)}$ |
| 64 | $55.1_{(4.8)}$ | $67.9_{(1.0)}$ | $56.9_{(5.7)}$ | $76.9_{(1.4)}$ | $60.4_{(5.3)}$ | $74.2_{(3.2)}$ |
| 512 | $55.1_{(5.1)}$ | $77.9_{(0.8)}$ | $55.3_{(6.1)}$ | $83.5_{(0.3)}$ | $67.7_{(2.0)}$ | $82.6_{(1.2)}$ |

| | RTE | | MRPC | | QQP | |
|---|---|---|---|---|---|---|
| $k$-shot | Lin. | FT | Lin. | FT | Lin. | FT |
| 0 | —— 53.1 —— | | —— 41.7 —— | | —— 42.7 —— | |
| 16 | $55.4_{(6.7)}$ | $57.6_{(6.3)}$ | $57.7_{(11.6)}$ | $68.9_{(2.4)}$ | $57.5_{(10.3)}$ | $61.7_{(6.5)}$ |
| 64 | $59.6_{(2.9)}$ | $67.3_{(2.7)}$ | $64.2_{(2.2)}$ | $73.8_{(1.7)}$ | $61.7_{(9.4)}$ | $72.7_{(1.8)}$ |
| 512 | $55.5_{(5.6)}$ | $75.4_{(3.0)}$ | $57.9_{(6.4)}$ | $84.8_{(0.8)}$ | $66.4_{(1.5)}$ | $78.7_{(0.8)}$ |

Table 4: Accuracies of pre-trained model (0-shot), linearized model (Lin., see Definition 3.1) and fine-tuned model (FT). Tasks that exhibit the Linearization property of kernel behavior (Definition 3.1) during fine-tuning will show that Lin. performance recovers a substantial amount of the gain in performance achieved by performing fine-tuning. Accuracies are averaged across 5 fine-tuning seeds for each value of $k$ and measured on the test set. This table corresponds to the bar chart in Figure 2.

neural networks, but it is not believed to be the full answer to neural networks in practice. In particular, kernel behavior implies the feature of the neural networks remains unchanged in the overparameterized setting, which is not true in practical pre-training of large models. In contrast,

| $k$-shot | SST-2 | MR | CR | MPQA | Subj | trec |
|---|---|---|---|---|---|---|
| 16 | $0.44_{(0.14)}$ | $0.409_{(0.14)}$ | $0.419_{(0.19)}$ | $0.414_{(0.08)}$ | $0.429_{(0.28)}$ | $2.023_{(0.8458)}$ |
| 64 | $0.41_{(0.07)}$ | $0.448_{(0.22)}$ | $0.416_{(0.05)}$ | $0.416_{(0.13)}$ | $0.601_{(0.17)}$ | $1.588_{(0.2295)}$ |

| $k$-shot | MNLI | SNLI | QNLI | RTE | MRPC | QQP |
|---|---|---|---|---|---|---|
| 16 | $0.666_{(0.16)}$ | $0.658_{(0.07)}$ | $0.482_{(0.10)}$ | $0.587_{(0.22)}$ | $0.632_{(0.21)}$ | $0.458_{(0.10)}$ |
| 64 | $0.859_{(0.16)}$ | $0.649_{(0.03)}$ | $0.582_{(0.12)}$ | $0.573_{(0.06)}$ | $0.857_{(0.08)}$ | $0.559_{(0.10)}$ |

Table 5: Average element-wise relative distance of $\mathcal{K}^{(\text{SGD})}$ computed on the pre-trained and best fine-tuned model. A smaller value indicates a higher likelihood that the Fixed Features property of kernel behavior (Definition 3.1) holds when performing fine-tuning. Distances are averaged across 5 seeds for each value of $k$ and measured on the LM-BFF test set (Gao et al., 2021).

| $k$-shot | Method | SST-2 | | MR | | QNLI | | QQP | |
|---|---|---|---|---|---|---|---|---|---|
| | | Standard | Prompt | Standard | Prompt | Standard | Prompt | Standard | Prompt |
| 16 | SGD-FT | $77.6_{(4.3)}$ | $\mathbf{89.0}_{(1.5)}$ | $64.8_{(5.2)}$ | $\mathbf{83.2}_{(2.4)}$ | $51.6_{(1.8)}$ | $62.1_{(3.1)}$ | $57.0_{(4.6)}$ | $\mathbf{62.1}_{(2.3)}$ |
| | Adam-FT | $\mathbf{78.1}_{(4.2)}$ | $88.3_{(1.2)}$ | $\mathbf{69.0}_{(6.0)}$ | $81.3_{(6.1)}$ | $\mathbf{56.7}_{(3.6)}$ | $\mathbf{63.1}_{(3.5)}$ | $\mathbf{58.5}_{(5.6)}$ | $61.8_{(4.5)}$ |
| 64 | SGD-FT | $85.6_{(1.9)}$ | $\mathbf{89.7}_{(0.4)}$ | $83.4_{(1.7)}$ | $85.6_{(1.1)}$ | $65.8_{(4.2)}$ | $72.8_{(2.2)}$ | $64.5_{(3.7)}$ | $69.2_{(1.3)}$ |
| | Adam-FT | $\mathbf{86.1}_{(1.2)}$ | $89.3_{(0.7)}$ | $\mathbf{83.9}_{(1.9)}$ | $\mathbf{86.0}_{(0.4)}$ | $\mathbf{71.5}_{(4.5)}$ | $\mathbf{74.2}_{(3.2)}$ | $\mathbf{65.0}_{(3.6)}$ | $\mathbf{69.8}_{(0.6)}$ |
| 512 | SGD-FT | $91.0_{(0.4)}$ | $\mathbf{91.8}_{(0.4)}$ | $\mathbf{88.7}_{(0.5)}$ | $\mathbf{89.0}_{(0.6)}$ | $81.6_{(1.5)}$ | $82.5_{(0.5)}$ | $75.8_{(0.6)}$ | $\mathbf{76.0}_{(1.0)}$ |
| | Adam-FT | $\mathbf{91.4}_{(0.7)}$ | $90.7_{(1.2)}$ | $88.4_{(0.8)}$ | $88.6_{(0.6)}$ | $\mathbf{82.2}_{(0.3)}$ | $\mathbf{82.6}_{(1.2)}$ | $\mathbf{76.1}_{(0.8)}$ | $75.7_{(0.9)}$ |

Table 6: Fine-tuning performance in the standard fine-tuning setting, where the contextual embedding of the `[CLS]` token is used for classification, and the prompt-based fine-tuning setting, where a prompt is added and the embedding for the `[MASK]` token is used (see Section 3). This table relates to Figure 1 by comparing the SGD fine-tuning results to the more common fine-tuning with Adam.

| $k$-shot | Method | SST-2 | MR | CR | QNLI | QQP |
|---|---|---|---|---|---|---|
| 16 | SGD-FT | $89.0_{(1.5)}$ | $83.2_{(2.4)}$ | $93.3_{(0.2)}$ | $62.1_{(3.1)}$ | $62.1_{(2.3)}$ |
| | SGD-LoRA FT | $89.1_{(0.6)}$ | $82.7_{(2.0)}$ | $92.6_{(0.8)}$ | $57.1_{(3.3)}$ | $59.8_{(3.0)}$ |
| | $\mathcal{K}^{(\text{SGD})}$ | $88.3_{(0.3)}$ | $84.7_{(1.5)}$ | $89.5_{(0.5)}$ | $60.1_{(3.3)}$ | $58.2_{(0.9)}$ |
| | $\mathcal{K}^{(\text{SGD})}_{\text{LoRA}}$ | $88.1_{(0.4)}$ | $84.9_{(1.4)}$ | $93.0_{(1.1)}$ | $59.2_{(3.6)}$ | $56.9_{(4.4)}$ |
| 64 | SGD-FT | $89.7_{(0.4)}$ | $85.6_{(1.1)}$ | $94.3_{(0.5)}$ | $72.8_{(2.2)}$ | $69.2_{(1.3)}$ |
| | SGD-LoRA FT | $90.0_{(0.2)}$ | $85.7_{(1.2)}$ | $93.9_{(0.7)}$ | $73.8_{(2.7)}$ | $68.3_{(2.4)}$ |
| | $\mathcal{K}^{(\text{SGD})}$ | $89.2_{(1.0)}$ | $86.4_{(0.6)}$ | $89.8_{(0.3)}$ | $67.3_{(1.6)}$ | $66.4_{(1.7)}$ |
| | $\mathcal{K}^{(\text{SGD})}_{\text{LoRA}}$ | $89.2_{(0.7)}$ | $85.6_{(1.6)}$ | $93.6_{(0.4)}$ | $66.0_{(1.6)}$ | $63.9_{(4.5)}$ |

Table 7: Accuracies of prompt-based SGD FT and prompt-based SGD-LoRA FT, along with their kernel analogs $\mathcal{K}^{(\text{SGD})}$ and $\mathcal{K}^{(\text{SGD})}_{\text{LoRA}}$, on a subset of tasks. SGD FT and SGD-LoRA FT achieve comparable performance, and $\mathcal{K}^{(\text{SGD})}$ and $\mathcal{K}^{(\text{SGD})}_{\text{LoRA}}$ also achieve comparable performance to each other. These experiments support Theorem 6.3.

Yang & Hu (2021) shows how the initialization variance, multiplier, and learning rate for each parameter can move training from the kernel behavior to the feature learning behavior. They further developed the Maximal Update Parametrization (abbreviated MUP or $\mu$P) where every parameter is updated maximally (in terms of scaling with width) while keeping the network stable. Yang et al. (2022) then extends $\mu$P to Transformers with Adam optimization, and showed empirically that for pre-training of large language models using $\mu$P, the optimal hyperparameters remain the same when increasing width. It allows more comprehensive hyperparameter searches on a smaller model and direct transfer of the resulting optimal hyperparameters to the larger model, thus providing better performance of pre-training.

In this section, we formally describe the right parametrization for kernel behavior in various fine-tuning settings. In general, we consider the overparameterized setting in which the width of the network goes to infinity. Additionally, we assume that when initializing a weight matrix of the model,

| Model size | SST-2 | MR | CR | MPQA | Subj | QNLI | RTE | MRPC | QQP |
|---|---|---|---|---|---|---|---|---|---|
| Base ($n = 768$) | 0.32 | 0.32 | 0.26 | 0.38 | 0.43 | 0.48 | 0.48 | 0.56 | 0.49 |
| Large ($n = 1024$) | 0.32 | 0.25 | 0.25 | 0.40 | 0.46 | 0.48 | 0.47 | 0.52 | 0.52 |

Table 8: We measure $\chi$ (Definition 3.2) in the prompt-based FT setting for RoBERTa-base and RoBERTa-large. A decrease in the $\chi$ value when going from RoBERTa-base to RoBERTa-large indicates the task may be solvable in the infinite-width limit (Definition 7.3). We find that for most tasks the eNTK can solve (Table 1), $\chi$ decreases as model width grows.

each entry of the matrix is drawn from i.i.d. Gaussian distribution. We use Tensor Programs (Yang, 2020b) as the framework for this setting.

## C.1 PRELIMINARIES

**Notations** Let $\xi \in \mathbb{R}^{d_{in}}$ be the input of the network. Let $n$ be the hidden dimension of the network and $d_{out}$ be the output dimension of the network. We define the network as a function of the following form:
$$f(\xi; \{U^i\}_i, \{W^j\}_j, V) = V^\top h(\xi; \{U^i\}_i, \{W^j\}_j),$$
where $\xi$ is the input, $U^i \in \mathbb{R}^{n \times d_{in}}$ are the input weight matrices, $W^j \in \mathbb{R}^{n \times n}$ are hidden weight matrices, $V \in \mathbb{R}^{n \times d_{out}}$ is the output weight matrix, and $h(\xi; \{U^i\}_i, \{W^j\}_j) \in \mathbb{R}^n$ is the input of last layer (readout layer). [4] We write $\mathcal{M}$ as the set of weight matrices, i.e., $\mathcal{M} = \{U^i\}_i \cup \{W^j\}_j \cup \{V\}$. For $M \in \mathcal{M}$, let $\nabla_M f(\xi)$ be the gradient of $f$ w.r.t. $M$ at input $\xi$.

For simplicity of the notations, we assume $d_{in} = 1$ in this section. Any non-trivial extension to $d_{in} > 1$ of results below will be noted along. For any weight matrix $M \in \mathcal{M}$, let $\gamma_M$ be the multiplier of $M$, such that $M$ is multiplied by $\gamma_M$ before performing matrix multiplication. Let $\eta_M$ be the learning rate of the weight $m$. Let $\sigma_M^2$ be the variance of entries of $m$ at initialization, so each entry of $m$ is drawn $\mathcal{N}(0, \sigma_M^2)$ independently.

Because we are considering the infinite-width limit, $f(\xi; \{U^i\}_i, \{W^j\}_j, V)$ actually represents a series of networks $\{f^n(\xi; \{U^{i,n}\}_i, \{W^{j,n}\}_j, V^n)\}_{n>0}$ of the same architecture, but $f^n$ has a hidden dimension $n$. When we say model $f$, it includes not only the information of the architecture, but also $\gamma_M, \eta_M, \sigma_M$ for every weight matrix $M$ in $f$ and the training optimizer of $f$.

Let $M_t$ be the weight matrix at time step $t$ of training. If the network is pre-trained, we let $M_{-1}$ be the weight matrix before pre-training, and $M_0$ be the parameters right after pre-training. Let $\Delta M_t = M_t - M_{t-1}$ be the change of each training step. Let $f_t$ be the network at step $t$ that
$$f_t(\xi) = f(\xi; \{U^i_t\}_i, \{W^j_t\}_j, V_t).$$

Let $\xi_t, y_t$ be the training input and target at step $t$, and let the loss function at step $t$ be $\mathcal{L}_t(f_t(\xi_t)) = \mathcal{L}(f_t(\xi_t), y_t)$. Let $\chi_t = \mathcal{L}'_t(f_t(\xi_t))$ be the derivative of the loss function.

**Big-O Notation** For a series of scalar random variables $c = \{c^n\}_{n>0}$ and a function $e : \mathbb{N} \to \mathbb{R}$, we say $c = \Theta(e(n))$ if there exist $A, B$ such that for sufficiently large $n$, $|c^n| \in [Ae(n), Be(n)]$ almost surely. For a series of vector random variables $x = \{x^n\}_{n>0}$, we say that $x$ is coordinate-wise $\Theta(n^a)$, or $x = \Theta(e(n))$ if this series of scalar random variables $\{\|x^n\|_2/\sqrt{n}\}_{n>0}$ is $\Theta(e(n))$. Similarly for the notation $O(e(n))$, $\Omega(e(n))$, and $o(e(n))$. For convenience, we assume every $e(n)$ in this section is equal to $n^a$ for some $a$.

**Tensor Programs** We refer reader to see Section 7 of Yang & Hu (2021) for detailed explanation and full definition of Tensor Programs. Here, we provide a simple overview of Tensor Programs:

**Definition C.1** (Definition 7.1 of Yang & Hu (2021))**.** A Tensor Program is a sequence of $\mathbb{R}^n$-vectors and $\mathbb{R}$-scalars inductively generated via one of the following ways from an initial set $\mathcal{C}$ of random scalars, $\mathcal{V}$ of random $\mathbb{R}^n$ vectors, and a set $\mathcal{W}$ of random $\mathbb{R}^{n \times n}$ matrices.

---

[4] We are able to describe transformers (without weight tying) in the definition. The bias can be regarded as input weights assuming there is a coordinate in $\xi$ that is always 1.

**MatMul** Given $W \in \mathbb{R}^{n \times n}$ and $x \in \mathbb{R}^n$, we can generate $Wx \in \mathbb{R}^n$ or $W^\top x \in \mathbb{R}^n$.

**Nonlin** Given $\phi : \mathbb{R}^k \times \mathbb{R}^l \to \mathbb{R}$, previous scalar $\theta_1, \ldots, \theta_l \in \mathbb{R}$ and vector $x^1, \ldots, x^k \in \mathbb{R}^n$, we can generate a new vector

$$\phi(x^1, \ldots, x^k; \theta_1, \ldots, \theta_l) \in \mathbb{R}^n$$

where $\phi(-; \theta_1, \ldots, \theta_l)$ applies coordinate-wise to each "$\alpha$-slice" $(x^1_\alpha, \ldots, x^k_\alpha)$.

**Moment** Given the same setup as above, we can also generate a new scalar

$$\frac{1}{n} \sum_{\alpha=1}^n \phi(x^1_\alpha, \ldots, x^k_\alpha; \theta_1, \ldots, \theta_l) \in \mathbb{R}.$$

Yang (2019; 2020a); Yang & Littwin (2021); Yang et al. (2022) show that Tensor Programs can express the computation, SGD/Adam optimization, and the kernel of practically any architecture.

The key result of the Tensor Programs is that we can represent the coordinates of any vector $x$ in the Tensor Program with a random variable $Z^x$, and represent any scalar $\theta$ with a deterministic scalar $\mathring{\theta}$. There is a way to define all $\mathring{\theta}$ and $Z^x$ correspond to the Tensor Program (cf. Definition 7.3 in Yang & Hu (2021)), and the Master Theorem of the Tensor Program shows that $\theta \to \mathring{\theta}$ when $n \to \infty$ (cf. Theorem 7.4 in Yang & Hu (2021)).

Although it is in general hard to compute $Z^x$ and $\mathring{\theta}$, it allows us to reason about the scales of vectors in the training of a network.

**Assumptions Related to Tensor Programs.** Since we are studying the infinite width limit and using Tensor Programs as our framework, there are some mild assumptions that we need in order to apply Tensor Programs and results in Yang & Hu (2021).

**Assumption C.2.** We assume the nework $f$ satisfies the following

a) The forward pass of $f$ in the infinite-width limit can be written as Tensor Programs.

b) The hidden vectors have $\Theta(1)$ coordinates at initialization.

c) The hidden vectors have $O(1)$ coordinates during training.

d) For any training scheme[5] and any constant $t$ and any input $\xi$, $f_t(\xi) = O(1)$.

e) There exist a training scheme and some constant $t$ and input $\xi$ such that $f_t(\xi) - f_0(\xi) = \Theta(1)$.

f) The activation function of $f$ is tanh or $\sigma$-gelu for a small enough $\sigma$ (so it approximates ReLU), where

$$\sigma\text{-}gelu(x) = \frac{1}{2} x \operatorname{erf}(\sigma^{-1} x) + \sigma \frac{e^{-\sigma^{-2} x^2}}{2\sqrt{\pi}} + \frac{x}{2}.$$

Furthermore, we have one assumption on SignGD:

g) SignGD is approximated as the sign function being replaced with $\epsilon$-sign for small enough $\epsilon$ when updating parameters, where $\epsilon\text{-sign}(x) = \frac{x}{|x|+\epsilon}$ is smoothed version of sign. We assume using different $\epsilon$ when computing the sign of $\nabla_M f$, so that $\epsilon$ for $\nabla_M f$ match the maximum scale of $\nabla_M f$.

b), c), d) and e) in Assumption C.2 together recover the definition of nontrivial stable network in Yang & Hu (2021). b) and c) ensure that the pre-activations in the network are not too large, so that the activation function like tanh are not trivialized to always output $\pm 1$. b) ensures that the pre-activations in the network are not too small at initialization, so the activation function is not trivialized to its first-order Taylor expansion. d) ensures the network output is bounded. e) ensures that the network is not frozen during training.

---

[5]Training scheme means a sequence of training examples $\{(\xi_t, y(\xi_t)\}$, and loss function $\ell(f_t(\xi_t), y(\xi_t))$.

f) and g) in Assumption C.2 assures all non-linear functions that appear in the Tensor Programs is pseudo-Lipschitz, which is required for the Master Theorem of Tensor Programs. g) also assures that $\epsilon$-sign is not trivialize to $0$ or sign when $\nabla_M f \neq \Theta(1)$.

## C.2   SignGD Kernel Derivation

**Definition C.3** (Formal Definition of Kernel Behavior). We say that this network training process demonstrates *kernel behavior* if the following properties are satisfied.

1. *Linearization*: The change of the network can be approximated by its first order Taylor expansion, i.e.,

$$\lim_{n \to \infty} \frac{f_t(\xi) - f_{t-1}(\xi)}{\chi_t} = \lim_{n \to \infty} \sum_{M \in \mathcal{M}} \left\langle \nabla_M f_{t-1}(\xi), \frac{\Delta M_t}{\chi_t} \right\rangle ;$$

2. *Fixed Features*: The gradients at step $t$ are approximately the same as before training, i.e.,

$$\forall M \in \mathcal{M}, \lim_{n \to \infty} \frac{\|\nabla_M f_t(\xi) - \nabla_M f_0(\xi)\|_2^2}{\max_{\xi'} \|\nabla_M f_0(\xi')\|_2^2} = 0.$$

Note that we define Linearization with both LHS and RHS divided by $\chi_t$ so it is meaningful for the case of $\chi_t = o(1)$. We do the same thing in the following theorem.

**Theorem C.4** (SignGD Kernel). *If SignGD training of $f$ demonstrates kernel behavior, then under Assumption C.2,*

$$\lim_{n \to \infty} \frac{f_t(\xi) - f_{t-1}(\xi)}{\chi_t} = \lim_{n \to \infty} \sum_{M \in \{U^i\}_i \cup \{W^j\}_j \cup \{V\}} -\eta_M \left\langle \nabla_M f_0(\xi), \epsilon\text{-sign}(\nabla_M f_0(\xi_t)) \right\rangle .$$

Note if $\eta_M = \eta$, the RHS of the equation above equals to

$$-\eta \langle \nabla f_0(\xi), \epsilon\text{-sign}(\nabla f_0(\xi_t)) \rangle \approx -\eta \mathcal{K}^{(\text{A-SignGD})}(\xi, \xi_t),$$

where the approximation comes from the difference between $\epsilon$-sign and sign.

*Proof.* By the update rule of SignGD, $\frac{\Delta M_t}{\chi_t} = -\eta_M \epsilon\text{-sign}(\nabla_M f_{t-1})$. It suffices to prove

$$\eta_M \left\langle \nabla_M f_t(\xi), \epsilon\text{-sign}(\nabla_M f_t(\xi_t)) \right\rangle = \eta_M \left\langle \nabla_M f_0(\xi), \epsilon\text{-sign}(\nabla_M f_0(\xi_t)) \right\rangle$$

when $n \to \infty$.

Since

$$\eta_M \left\langle \nabla_M f_t(\xi), \epsilon\text{-sign}(\nabla_M f_t(\xi_t)) \right\rangle - \eta_M \left\langle \nabla_M f_0(\xi), \epsilon\text{-sign}(\nabla_M f_0(\xi_t)) \right\rangle$$

$$= \eta_M \left\langle \nabla_M f_t(\xi) - \nabla_M f_0(\xi), \epsilon\text{-sign}(\nabla_M f_t(\xi_t)) \right\rangle + \tag{4}$$

$$\eta_M \left\langle \nabla_M f_t(\xi), \epsilon\text{-sign}(\nabla_M f_t(\xi_t)) - \epsilon\text{-sign}(\nabla_M f_0(\xi_t)) \right\rangle + \tag{5}$$

$$\eta_M \left\langle \nabla_M f_t(\xi) - \nabla_M f_0(\xi), \epsilon\text{-sign}(\nabla_M f_t(\xi_t)) - \epsilon\text{-sign}(\nabla_M f_0(\xi_t)) \right\rangle , \tag{6}$$

we only need to prove Equations (4) to (6) are all 0 when $n \to \infty$.

Let $\xi^* = \arg\max_{\xi^*} \|\nabla_M f_0(\xi')\|_2^2$ be the input of maximum gradient scale, then by Fixed Features, we have

$$\frac{\|\nabla_M f_t(\xi) - \nabla_M f_0(\xi)\|_2}{\|\nabla_M f_0(\xi^*)\|_2} = o(1).$$

Since $\epsilon\text{-sign}(x) - \epsilon\text{-sign}(y) \leq |x - y|/\epsilon$,

$$\|\epsilon\text{-sign}(\nabla_M f_t(\xi)) - \epsilon\text{-sign}(\nabla_M f_0(\xi))\|_2 \leq \|\nabla_M f_t(\xi) - \nabla_M f_0(\xi)\|_2/\epsilon. \tag{7}$$

Combined with $\|\nabla_M f_0(\xi^*)\|_2/\sqrt{N} = \Theta(\epsilon)$ ($N$ is the number of entries of $M$, this is by g) of Assumption C.2), we have

$$
\begin{aligned}
&\frac{\|\epsilon\text{-sign}(\nabla_M f_t(\xi)) - \epsilon\text{-sign}(\nabla_M f_0(\xi))\|_2}{\|\epsilon\text{-sign}(\nabla_M f_0(\xi^*))\|_2} \\
\leq\ &\frac{\|\nabla_M f_t(\xi) - \nabla_M f_0(\xi)\|_2/\epsilon}{\|\epsilon\text{-sign}(\nabla_M f_0(\xi^*))\|_2} \qquad\qquad \text{by eq. (7)} \\
=\ &\frac{\|\nabla_M f_t(\xi) - \nabla_M f_0(\xi)\|_2}{\|\nabla_M f_0(\xi^*)\|_2} \cdot \frac{\|\nabla_M f_0(\xi^*)\|_2/\sqrt{N}}{\epsilon\|\epsilon\text{-sign}(\nabla_M f_0(\xi^*))\|_2/\sqrt{N}} \\
=\ &\frac{\|\nabla_M f_t(\xi) - \nabla_M f_0(\xi)\|_2}{\|\nabla_M f_0(\xi^*)\|_2} \cdot \Theta(1) = o(1).
\end{aligned}
$$

By d) in Assumption C.2, and consider the training scheme that sets $\xi_1 = \xi^*$ and the loss function $\ell$ so $\chi_1 = \Theta(1)$, then

$$
\frac{f_1(\xi^*) - f_0(\xi^*)}{\chi_1} = -\eta_M \langle \nabla_M f_0(\xi^*), \epsilon\text{-sign}(\nabla_M f_0(\xi^*)) \rangle = O(1).
$$

And it is easy to see that $\langle \nabla_M f_0(\xi^*), \epsilon\text{-sign}(\nabla_M f_0(\xi^*)) \rangle$ and $\|\nabla_M f_0(\xi^*)\|_2\|\epsilon\text{-sign}(\nabla_M f_0(\xi^*))\|_2$ has the same scale.

Now it suffices to prove Equations (4) to (6) divided by $\eta_M\|\nabla_M f_0(\xi^*)\|_2\|\epsilon\text{-sign}(\nabla_M f_0(\xi^*))\|_2$ are all 0 when $n \to \infty$. For Equation (4),

$$
\begin{aligned}
&\frac{\eta_M \langle \nabla_M f_t(\xi) - \nabla_M f_0(\xi), \epsilon\text{-sign}(\nabla_M f_t(\xi_t)) \rangle}{\eta_M\|\nabla_M f_0(\xi^*)\|_2\|\epsilon\text{-sign}(\nabla_M f_0(\xi^*))\|_2} \\
\leq\ &\frac{\|\nabla_M f_t(\xi) - \nabla_M f_0(\xi)\|_2\|\epsilon\text{-sign}(\nabla_M f_t(\xi_t))\|_2}{\|\nabla_M f_0(\xi^*)\|_2\|\epsilon\text{-sign}(\nabla_M f_0(\xi^*))\|_2} \\
=\ &\frac{\|\nabla_M f_t(\xi) - \nabla_M f_0(\xi)\|_2}{\|\nabla_M f_0(\xi^*)\|_2} = o(1).
\end{aligned}
$$

Similarly, for Equation (5),

$$
\begin{aligned}
&\frac{\eta_M \langle \nabla_M f_t(\xi), \epsilon\text{-sign}(\nabla_M f_t(\xi_t)) - \epsilon\text{-sign}(\nabla_M f_0(\xi_t)) \rangle}{\eta_M\|\nabla_M f_0(\xi^*)\|_2\|\epsilon\text{-sign}(\nabla_M f_0(\xi^*))\|_2} \\
\leq\ &\frac{\|\epsilon\text{-sign}(\nabla_M f_t(\xi)) - \epsilon\text{-sign}(\nabla_M f_0(\xi))\|_2}{\|\epsilon\text{-sign}(\nabla_M f_0(\xi^*))\|_2} = o(1),
\end{aligned}
$$

and for Equation (6),

$$
\begin{aligned}
&\frac{\eta_M \langle \nabla_M f_t(\xi) - \nabla_M f_0(\xi), \epsilon\text{-sign}(\nabla_M f_t(\xi_t)) - \epsilon\text{-sign}(\nabla_M f_0(\xi_t)) \rangle}{\eta_M\|\nabla_M f_0(\xi^*)\|_2\|\epsilon\text{-sign}(\nabla_M f_0(\xi^*))\|_2} \\
\leq\ &\frac{\|\epsilon\text{-sign}(\nabla_M f_t(\xi)) - \epsilon\text{-sign}(\nabla_M f_0(\xi))\|_2}{\|\epsilon\text{-sign}(\nabla_M f_0(\xi^*))\|_2} \cdot \frac{\|\nabla_M f_t(\xi) - \nabla_M f_0(\xi)\|_2}{\|\nabla_M f_0(\xi^*)\|_2} = o(1).
\end{aligned}
$$

$\square$

## C.3 PROMPT-BASED FINE-TUNING

In this section, we prove that prompt-based fine-tuning exhibits kernel behavior with the assumption of $\chi_t = o(1)$. Prompt-based fine-tuning is trained directly on the pre-trained network without substituting or adding any parameters. Without our assumption, it is obvious that the behavior of fine-tuning and pre-training is the same from the perspective of the Tensor Programs.

**Theorem C.5.** *If the downstream task is solvable for network $f$, that is, for any $t$, $\chi_t = o(1)$, then under Assumption C.2, the fine-tuning of $f$ exhibits kernel behavior (Definition C.3).*

Below we provide a proof that is heavily based on Tensor Programs and the analysis in Yang & Hu (2021). For readers who are not familiar with Tensor Programs, we provide intuitive examples in the next few subsections, where we focus on a three-layer linear network parameterized with $\mu$P.

*Proof.* We first prove the theorem under the assumption that the network is a multilayer perceptron and the optimizer is SGD, which is the same setting as Yang & Hu (2021). Then we will extend to more general cases.

Consider the following $L$-hidden-layer perceptron:

$$h^1(\xi) = U\xi,$$

and

$$x^l(\xi) = \phi(h^l(\xi)), \quad h^{l+1}(\xi) = W^{l+1}x^l(\xi), \text{ for } l = 1, \ldots, L-1,$$

and

$$f(\xi) = Vx^L(\xi).$$

Following Yang & Hu (2021), we let the learning rate for every parameter equal to $\eta n^{-c}$. Let $W^1 = U$ and $W^{L+1} = V$, and for $l = 1, \ldots, L+1$, we parametrize $W^l$ as $W^l = \gamma_l w^l$ for actual trainable parameter $w^l$, and we initialize each coordinate $w^l$ i.i.d. from $\mathcal{N}(0, \sigma_l^2)$. The setting covers all parameterizations based on Lemma C.6. For convenience, we assume $\gamma_l = n^{-a_l}$ and $\sigma_l = n^{-b_l}$. Without loss of generality, we further assume that $\chi_t = \Theta(n^{-d})$ (rather than $\chi_t = o(1)$ only).

By Theorem 3.3 of Yang & Hu (2021), stable network implies

$$r \triangleq \min(a_{L+1} + b_{L+1}, 2a_{L+1} + c) + c - 1 + \min_{l=1}^{L}[2a_l + \mathbb{I}(l=1)] \geq 0.$$

Also by Theorem 3.8 of Yang & Hu (2021), for nontrivial stable network (included in Assumption C.2), if $r > 0$ then there exists a kernel $\mathcal{K}$ such that

$$f_{t+1}(\xi) = f_t(\xi) - \eta\chi_t\mathcal{K}(\xi, \xi_t),$$

which is very close to our definition of kernel behavior. In fact, we will prove that they are equivalent in the fine-tuning case.

Since $\chi_t = \Theta(n^{-d})$ for fine-tuning, it is equivalent to set the learning rate to $\eta n^{-c-d}$ and replace $\chi_t$ with $\hat{\chi}_t = n^d\chi_t$. Formally, we are considering the following training scheme: at the pre-training stage, $r \geq 0$ (so it could demonstrate feature learning or kernel behavior); at the fine-tuning stage, $c$ is increased to $c' \triangleq c + d > c$, thus, the corresponding $r$ is increased to be strictly greater than 0. Therefore, it suggests kernel behavior with following caveats.

**Caveat 1: do pre-training affect the result?** The answer is *effectively NO*. First of all, the scale of the update on $W^l$, $h^l$, $x^l$ and $f$ are all multiplied by $n^{-d}$ when switching from the pre-training stage ($\eta n^{-c}$ learning rate) to the fine-tuning stage($\eta n^{-c-d}$ learning rate). The scales are exactly the same as training from scratch with $\eta n^{-c-d}$ learning rate except $b_{L+1}$ needs to be changed to $b'_{L+1} \triangleq \min(b_{L+1}, a_{L+1} + c)$. Note this change of $b_{L+1}$ does not affect the fact that $r$ is updated to $r' \triangleq r + d > 0$.

**Caveat 2: does $r' > 0$ formally implies our definition of kernel behavior (Definition C.3)?** The answer is *YES*. We first prove Fixed Features in Definition C.3. The gradient of matrix $W^l$ is equal to outer product between $\nabla_{h^l} f$ (gradient w.r.t. $h^l$) and $x^{l-1}$. Let $dh_t^l$ be the normalized gradient w.r.t. $h^l$ at step $t$ (so $dh_t^l = \Theta(1)$), and $x_t^l$ be the $x^l$ at step $t$ ($x_t^l = \Theta(1)$ without normalization). It suffices to prove $dh_t^l - dh_0^l = O(1)$ and $x_t^l - x_0^l = o(1)$. The later was proved by Proposition H.27 of Yang & Hu (2021). To prove $dh_t^l - dh_0^l = O(1)$, we let $dx_t^l$ be the the normalized gradient w.r.t. $x^l$ at step $t$, and compute the scale of $dh_t^l - dh_{t-1}^l$ and $dx_t^l - dx_{t-1}^l$ inductively from $l = L$ to $l = 1$. We obtain that they both has the same scale of

$$n^{-\min(2a_{L+1}+c-a_{L+1}-b'_{L+1}, a_{L+1}+b_{L+1}+c'-1+\min_{m=l+1}^{L}2a_m)} \leq n^{-\min(0, r')} = 1,$$

the inequality is because $b'_{L+1} \leq a_{L+1} + c$ and $r' \leq a_{L+1} + b_{L+1} + c' - 1 + \min_{m=l+1}^{L} 2a_m$.

Second, we prove Linearization in Definition C.3. We need to first make a very slight modification to the Tensor Program in Yang & Hu (2021), that is, changing the computation of $f_t(\xi) - f_{t-1}(\xi)$ to $n^{-d}(f_t(\xi) - f_{t-1}(\xi))$. By Theorem H.32 of Yang & Hu (2021) and its definition of $\Sigma$, we can show

that

$$\lim_{n \to \infty} n^{-d}(f_t(\xi) - f_{t-1}(\xi)) = \lim_{n \to \infty} \sum_{l=1}^{L+1} \eta n^{-c-d} \langle \nabla_{W^l} f_{t-1}(\xi), \nabla_{W^l} f_{t-1}(\xi_t) \rangle$$

$$= \lim_{n \to \infty} \sum_{l=1}^{L+1} \left\langle \nabla_{W^l} f_{t-1}(\xi), \frac{\Delta W_t^l}{n^{-d}} \right\rangle.$$

**From SGD to SignGD.** Since $\text{sign}(xy) = \text{sign}(x)\text{sign}(y)$, the update of matrix $W^l$ can still be written as outer product of two vectors, i.e., $\Delta W_t^l = \eta n^{-c} \chi_t \text{sign}(\nabla_{h^l} f_{t-1}) \otimes \text{sign}(x_{t-1}^{l-1})$. After applying sign, the scale of vector changes. If the parametrization is the same, the scales of vectors using SignGD will be different from those using SGD. This can be easily resolved by changing learning rates for each parameter, so the scaling change brought by sign is corrected. Furthermore, as mentioned in Assumption C.2, we need to approximate sign by a smoothed version $\epsilon$-sign so the Master Theorem of Tensor Programs can still stands.

**Extension to universal architecture.** The theorem is correct for any network whose first forward pass can be written as Tensor Programs. Given this condition, the forward pass, backward pass and kernel of any steps can be written as Tensor Programs (Yang, 2020a;b). To analyse the scaling of the Tensor Program will need the following steps:

1. *Extension to general computation graph.* We can still inductive reason about the scale of preactivations and activations by the topological order of the computation graph; and similarly reason about the gradient by the reverse topological order.

2. *Extension to weight sharing.* We may use weights multiple times in a forward pass. The preactivations, activations and their gradients will not be affected. Only the update of a weight is now sum of several vector outer product depending on the number of occurrence of the weight.

$\square$

### C.4 $\mu$P FOR SGD AND SIGNGD

In the following subsections, we provide more intuitions to Theorem C.5 and some other situations where kernel behavior exhibits. Although we care about all type of pre-trained models, we are mostly interested in models with feature learning behavior. For pre-trained models with kernel behavior, it is obvious that fine-tuning with the same settings as pre-training (corresponds to Prompt-based FT) will lead it to the kernel behavior. Furthermore, Theorem H.17 of Yang & Hu (2021) proved that if the last layer is replaced with a freshly initialized layer (corresponds to Standard FT), fine-tuning from a pre-trained models with kernel behavior is the same as training the downstream task from scratch.

Among all the models with feature learning behavior, $\mu$P is the most special one where each parameter itself (except the last layer) can push the model to learn feature. Therefore, we use $\mu$P as an example to give a proof of intuitive understanding.

The formulation of $\mu$P contains three sets of hyperparameters: initial variance of $M$, multiplier of $M$ and learning rate of $M$ for $M \in \{U^i\}_i \cup \{W^j\}_j \cup \{V\}$. However, even if we restrict these three hyperparameters to be in the form of $n^\alpha$, $\mu$P is not unique, because there is one degree of freedom for each weight according to the following lemma.

**Lemma C.6** (Lemma J.1 of Yang et al. (2022)). *Consider a weight matrix $M$ with learning rate $C$, initialized as $M \sim \mathcal{N}(0, B^2)$, and with a multiplier $A$. Then for any $\gamma > 0$, $f_t(\xi)$ stays fixed for all $t$ and $\xi$ if we set*

- $A \leftarrow A\gamma, B \leftarrow B/\gamma, C \leftarrow C/\gamma^2$ *if training with SGD.*

- $A \leftarrow A\gamma, B \leftarrow B/\gamma, C \leftarrow C/\gamma$ *if training with Adam.*

Note the conclusion about Adam in Lemma C.6 also extends to SignGD.

With Lemma C.6, we can always set the multiplier of any weight matrix $M$ to be 1, which leave us only the initialization variance $\sigma_M^2$ and learning rate $\eta_M$. Furthermore, in terms of the scale at initialization and the scale of updates, $\mu$P for SGD and SignGD are entirely the same. The only difference would be learning rate. We provide details in Table 9 (recall $M_{-1}$ is the weight $M$ at initialization of pre-training, $\Delta M_0 = M_0 - M_{-1}$ is the overall change of weight in pre-training).

Since we have different learning rate for $\eta_M$, the kernel that we care is defined as

$$\mathcal{K}(\xi, \xi') = \sum_{M \in \mathcal{M}} \eta_M \langle \nabla_W f(\xi), \phi(\nabla_W f(\xi')) \rangle,$$

where $\phi$ is identity if the algorithm is SGD, $\phi = \text{sign}$ if the algorithm is SignGD. And we want to prove the dynamic of the network follows

$$\frac{f_t(\xi) - f_{t-1}(\xi)}{\chi_t} \to -\mathcal{K}(\xi, \xi_t) \quad \text{when } n \to \infty.$$

### C.5 PROMPT-BASED FINE-TUNING: A LINEAR EXAMPLE

As an intuitive example, we consider a three-layer linear network

$$f(\xi; U, W, V) = V^\top W U \xi.$$

For simplicity, we train the network with SGD, and freeze $V$ so $\eta_V = 0$. Then we have $\nabla_U f = W^\top V \xi^\top$ and $\nabla_W f = V(U\xi)^\top$. We assume $|\langle \xi, \xi' \rangle| > 0$ for any $\xi, \xi'$.

**Zero step (Pre-training)** We model the pre-training of $f$ as one step of training with $\chi_0 = \Theta(1)$. Then we have $\Delta U_0 = -\eta_U \chi_0 W_{-1}^\top V \xi_0^\top$, and $\Delta W_0 = -\eta_W \chi_0 V(U_{-1}\xi_0)^\top$. Since $W_{-1}^\top$ is independent from $V$, we have $W_{-1}^\top V = \Theta(1/n)$, thus $\Delta U_0 = \Theta(1)$ matching Table 9. On the other hand, it is obvious that $\Delta W_0 = \Theta(1/n)$ because $V = \Theta(1/n)$ and $U = \Theta(1)$, also matching Table 9.

Then the function is now

$$
\begin{aligned}
f_0(\xi) &= V^\top (W_{-1} + \Delta W_0)(U_{-1} + \Delta U_0)\xi \\
&= V^\top (W_{-1} - \eta_W \chi_0 V(U_{-1}\xi_0)^\top)(U_{-1}\xi - \eta_U \chi_0 W_{-1}^\top V \langle \xi_0, \xi \rangle) \\
&= V^\top W_{-1} U_{-1}\xi - \eta_U \chi_0 \|W_{-1}^\top V\|_2^2 \langle \xi_0, \xi \rangle - \eta_W \chi_0 \|V\|^2 \langle U_{-1}\xi_0, U_{-1}\xi \rangle \\
&\quad + \eta_W \eta_U \chi_0^2 \|V\|^2 \langle \xi_0, \xi \rangle V^\top W_{-1} U_{-1}\xi.
\end{aligned}
$$

It is not difficult to see that $\eta_U \chi_0 \|W_{-1}^\top V\|_2^2 \langle \xi_0, \xi \rangle$, $\eta_W \chi_0 \|V\|^2 \langle U_{-1}\xi_0, U_{-1}\xi \rangle$, and $\eta_W \eta_U \chi_0^2 \|V\|^2 \langle \xi_0, \xi \rangle$ are all $\Theta(1)$. Unfortunately, here $V^\top W_{-1} U_{-1}\xi = 0$ in the infinite-width limit, but if we train one more step, it is easy to see that all four terms of $f_0$ is $\Theta(1)$.

**First step** At the first step of fintuning, we have $\Delta U_1 = -\eta_U \chi_1 W_0^\top V \xi_1^\top$ and $\Delta W_1 = -\eta_W \chi_1 V(U_0 \xi_1)^\top$. Then the function is now

$$f_1(\xi) = V^\top (W_0 + \Delta W_1)(U_0 + \Delta U_1)\xi,$$

and

$$f_1(\xi) - f_0(\xi) = V'^\top \Delta W_1 U_0 \xi + V^\top W_0 \Delta U_1 \xi + V^\top \Delta W_1 \Delta U_1 \xi. \tag{8}$$

| coordinate-wise scale | $M = U^i$ | $M = W^j$ | $M = V$ |
|:---:|:---:|:---:|:---:|
| $M_{-1}$ | $\Theta(1)$ | $\Theta(1/\sqrt{n})$ | $\Theta(1/n)$ |
| $\Delta M_0$ | $\Theta(1)$ | $\Theta(1/n)$ | $\Theta(1/n)$ |
| $\eta_M$ for SGD | $\Theta(n)$ | $\Theta(1)$ | $\Theta(1/n)$ |
| $\eta_M$ for signGD/Adam | $\Theta(1)$ | $\Theta(1/n)$ | $\Theta(1/n)$ |

Table 9: Scales of initialization, update and learning rate for $\mu$P in pre-training.

Note that the sum of the first and second terms is exactly $-\chi_1 \mathcal{K}(\xi, \xi_1)$.

Plug in $\Delta W_1 = -\eta_W \chi_1 V(U_0 \xi_1)^\top$ into the first term of eq. (8),

$$V^\top \Delta W_1 U_0 \xi = -\eta_W \chi_1 V^\top V(U_0\xi_1)^\top U_0 \xi = \Theta(\chi_1),$$

because

$$
\begin{aligned}
(U_0\xi_1)^\top U_0\xi &= (U_{-1}\xi_1 + \Delta U_0\xi_1)^\top (U_{-1}\xi + \Delta U_0\xi) \\
&= \langle U_{-1}\xi_1, U_{-1}\xi \rangle - \eta_U \chi_0 \langle \xi_1, \xi_0 \rangle f_{-1}(\xi) - \eta_U\chi_0\langle\xi,\xi_0\rangle f_{-1}(\xi_1) + \|\Delta U_0\|^2 \langle \xi_1, \xi \rangle \\
&= \Theta(n).
\end{aligned}
$$

Plug in $\Delta U_1 = -\eta_U\chi_1 W_0^\top V\xi_1^\top$ into the second term of eq. (8), we have

$$V^\top W_0 \Delta U_1 \xi = -\eta_U\chi_1 V^\top W_0 W_0^\top V\xi_1^\top \xi = \Theta(\chi_1)$$

because

$$
\begin{aligned}
V^\top W_0 W_0^\top V &= \|(W_{-1} + \Delta W_0)^\top V, (W_{-1} + \Delta W_0)^\top V\|_2^2 \\
&= \|W_{-1}^\top V\|_2^2 + \eta_W^2 \chi_0^2 \|V\|_2^4 \|U_{-1}\xi_0\|_2^2 - 2\eta_W\chi_0 \|V\|_2^2 f_{-1}(\xi_0) = \Theta(1/n).
\end{aligned}
$$

The third term of eq. (8) equals

$$\eta_U\eta_W\chi_1^2 V^\top V(U_0\xi_1)^\top W_0^\top V\xi_1^\top \xi = \eta_U\eta_W\chi_1^2 \|V\|^2 \langle \xi_1,\xi\rangle f_0(\xi_1) = \Theta(\chi_1^2).$$

Therefore, $\frac{f_1(\xi) - f_0(\xi)}{\chi_1} \to \mathcal{K}(\xi, \xi_1)$

**Second step** At the second step of fine-tuning, we have $\Delta U_2 = -\eta_U\chi_1 W_1^\top V\xi_2^\top$, and $\Delta W_2 = -\eta_W\chi_1 V(U_1\xi_2)^\top$ and

$$f_2(\xi) - f_1(\xi) = V^\top \Delta W_2 U_1 \xi + V^\top W_1 \Delta U_2 \xi + V^\top \Delta W_2 \Delta U_2 \xi. \tag{9}$$

Assuming $\chi_2$ and $\chi_1$ share the same order, then when $n \to \infty$,

$$
\begin{aligned}
\frac{f_2(\xi) - f_1(\xi)}{\chi_2} &= V^\top \Delta W_2 U_1\xi/\chi_2 + V^\top W_1 \Delta U_2 \xi/\chi_2 \\
&= -\eta_W V^\top V(U_1\xi_2)^\top U_1\xi - \eta_U V^\top W_1 W_1^\top V\xi_2^\top \xi \\
&= -\eta_W V^\top V(U_0\xi_2)^\top U_0\xi - \eta_U V^\top W_0 W_0^\top V\xi_2^\top \xi \\
&= -\mathcal{K}(\xi, \xi_2).
\end{aligned}
$$

**$t$th step** Same as the second step by noting $\Delta U_t, \Delta W_t$ always have smaller order than $\Delta U_0$ and $\Delta W_0$.

## C.6 STANDARD FINE-TUNING

In standard fine-tuning, $V$ is replaced with a randomly initialized matrix $V'$. That is, the model at the step $t$ of fine-tuning is $f_t(\xi) = f(\xi; \{U_t^i\}_i, \{W_t^j\}_j, V_t')$. We set $V_0' \sim \mathcal{N}(0, \frac{\sigma^2}{n} I_{n \times d_{out}})$, which has a larger scale than $V$ in $\mu$P. In this section, we will prove that this standard fine-tuning has kernel behavior.

We still consider a three-layer linear network

$$f(\xi; U, W, V') = V'^\top W U \xi$$

where $V' \in \mathbb{R}^{n \times 1}$ (i.e., $d_{out} = 1$) and we freeze $V'$ during the fine-tuning so it is not trained. Then we have $\nabla_U f = W^\top V'\xi^\top$ and $\nabla_W f = V'(U\xi)^\top$.

| coordinate-wise scale | $M = U^i$ | $M = W^j$ | $M = V$ | $M = V'$ |
|:---:|:---:|:---:|:---:|:---:|
| $M_{-1}$ | $\Theta(1)$ | $\Theta(1/\sqrt{n})$ | $\Theta(1/n)$ | - |
| $\Delta M_0$ | $\Theta(1)$ | $\Theta(1/n)$ | $\Theta(1/n)$ | $\Theta(1/\sqrt{n})$ |
| $\Delta M_t$ | $\Theta(1/\sqrt{n})$ | $\Theta(1/n\sqrt{n})$ | - | $\Theta(1/n)$ |
| $\eta_M$ for SGD | $\Theta(1)$ | $\Theta(1/n)$ | - | $\Theta(1/n)$ |
| $\eta_M$ for SignGD/Adam | $\Theta(1/\sqrt{n})$ | $\Theta(1/n\sqrt{n})$ | - | $\Theta(1/n)$ |

Table 10: Scales for standard fine-tuning w.r.t. $n$

**First step** At the first step of fintuning, we have $\Delta U_1 = -\eta_U \chi_1 W_0^\top V' \xi_1^\top$, and $\Delta W_1 = -\eta_W \chi_1 V' (U_0 \xi_1)^\top$. Then the function is now

$$f_1(\xi) = V'^\top (W_0 + \Delta W_1)(U_0 + \Delta U_1)\xi,$$

and

$$f_1(\xi) - f_0(\xi) = V'^\top \Delta W_1 U_0 \xi + V'^\top W_0 \Delta U_1 \xi + V'^\top \Delta W_1 \Delta U_1 \xi. \tag{10}$$

Note the first order and the second order term is exactly $-\chi_1 K(\xi, \xi_1)$.

Plug in $\Delta W_1 = -\eta_W \chi_1 V'(U_0 \xi_1)^\top$ into the first term of eq. (10),

$$V'^\top \Delta W_1 U_0 \xi = -\eta_W \chi_1 V'^\top V'(U_0 \xi_1)^\top U_0 \xi = \Theta(1)$$

since $(U_0 \xi_1)^\top U_0 \xi = \Theta(n)^6$.

Plug in $\Delta U_1 = -\eta_U \chi_1 W_0^\top V' \xi_1^\top$ into the second term of eq. (10), we have (assuming $|\xi_1^\top \xi| > 0$)

$$V'^\top W_0 \Delta U_1 \xi = -\eta_U \chi_1 V'^\top W_0 W_0^\top V' \xi_1^\top \xi = \Theta(1)$$

because $V'^\top W_0 W_0^\top V' = \Theta(1)$.[7]

It is easy to verify that the third term is $O(1/\sqrt{n})$. Therefore, $f_1(\xi) - f_0(\xi)$ converges to its first-order Talyer expansion when $n \to \infty$.

**Second step** At the second step of fine-tuning, we have $\Delta U_2 = -\eta_U \chi_1 W_1^\top V' \xi_2^\top$, and $\Delta W_2 = -\eta_W \chi_1 V'(U_1 \xi_2)^\top$ and

$$f_2(\xi) - f_1(\xi) = V'^\top \Delta W_2 U_1 \xi + V'^\top W_1 \Delta U_2 \xi + V'^\top \Delta W_2 \Delta U_2 \xi. \tag{11}$$

We remove all terms of $o(1)$, then

$$\begin{aligned}
f_2(\xi) - f_1(\xi) &\approx V'^\top \Delta W_2 U_1 \xi + V'^\top W_1 \Delta U_2 \xi \\
&= -\eta_W \chi_2 V'^\top V'(U_1 \xi_2)^\top U_1 \xi + -\eta_U \chi_2 V'^\top W_1 W_1^\top V' \xi_2^\top \xi \\
&\approx -\eta_W \chi_2 V'^\top V'(U_0 \xi_2)^\top U_0 \xi + -\eta_U \chi_2 V'^\top W_0 W_0^\top V' \xi_2^\top \xi \\
&= -\chi_2 K(\xi, \xi_2).
\end{aligned}$$

**$t$th step** Same as the second step by noting $\Delta U_t$, $\Delta W_t$ always have smaller order than $\Delta U_0$ and $\Delta W_0$.

### C.7 PROMPT-BASED FINE-TUNING WITH PROJECTION

If $\chi = \Theta(1)$ in Prompt-based FT, we need some modification in order to push the model to kernel regime, otherwise, the network will stay in feature learning regime or it will not learn (if we decrease learning rate). In particular, we add a randomized projection matrix before the last layer so the function is now equal to

$$f_t(\xi) = V_t^\top W_t' h(\xi; \{U_t^i\}_i, \{W_t^j\}_j) = V_t'^\top h(\xi; \{U_t^i\}_i, \{W_t^j\}_j)$$

where $V_t' \triangleq W_t'^\top V_t$, and $W_0' \sim \mathcal{N}(0, \sigma^2 I_{n \times n})$.

Now we consider the linear example again where

$$f(\xi; U, W, V, W') = V^\top W' W U \xi = V'^\top W U \xi.$$

If we freeze both $V$ and $W'$ during the fine-tuning and let $d_{out} = 1$, then it is equivalent to the linear example in Appendix C.6 with $V' \sim \mathcal{N}(0, \sigma^2 \|V\|_2^2 I_{n \times 1})$ (note $\|V\|_2^2 = \Theta(1/n)$).

---

[6]It is clear that $(U_0 \xi_1)^\top U_0 \xi = O(n)$. $(U_0 \xi_1)^\top U_0 \xi = o(n)$ would mean there is some weird correlation.

[7]It is easy to see that $\mathbb{E}[V'^\top W_0 W_0^\top V'] = \Theta(1)$ and $\mathrm{Var}(V'^\top W_0 W_0^\top V') = \Theta(\|W_0 W_0^\top\|_F^2 / n^2)$. By Yang & Hu (2021), $\|W_0 W_0^\top\|_F^2 / n^2 = o(1)$.

# D   SUBSPACE-BASED FINE-TUNING METHODS

We start by restating the Johnson-Lindenstrauss lemma, which preserves inner products under random projection.

**Lemma D.1** (Johnson-Lindenstrauss). *Let $u, v \in \mathbb{R}^d$ such that $\|u\| \leq 1$ and $\|v\| \leq 1$. Choose $k = 20 \log N/\epsilon^2$, where $N$ is the number of datapoints. Let $h = \frac{1}{\sqrt{k}} Ax$, where $A \in \mathbb{R}^{k \times d}$ with each entry sampled i.i.d. from $\mathcal{N}(0, 1)$ or $\mathcal{U}(-1, 1)$. Then,*

$$\Pr[|u \cdot v - h(u) \cdot h(v)| \geq \epsilon] \leq \exp(-\epsilon^2 k/4)$$

**Lemma D.2** (Norm Preservation (Johnson-Lindenstrauss)). *Let $x \in \mathbb{R}^n$ and assume the entries of $A \in \mathbb{R}^{k \times n}$ are sampled i.i.d. from $\mathcal{N}(0, 1)$. Then,*

$$\Pr\left[ (1 - \epsilon)\|x\|^2 \leq \left\| \frac{1}{\sqrt{k}} Ax \right\|^2 \leq (1 + \epsilon)\|x\|^2 \right] \geq 1 - \exp(-(\epsilon^2 - \epsilon^3)k/4)$$

**Lemma D.3** (Inner Product Preservation). *Let $u, v \in \mathbb{R}^k$ such that $\|u\|, \|v\| \leq B$. Assume $|\langle u, v \rangle| \geq c\|u\|\|v\|$ (i.e., $u$ and $v$ are not orthogonal). Choose $k = 20 \log N/\epsilon^2$, where $N$ is the number of datapoints. Let $h(x) = \frac{1}{\sqrt{k}} Ax$, where $A \in \mathbb{R}^{k \times n}$ with each entry sampled i.i.d. from $\mathcal{N}(0, 1)$. Then,*

$$\Pr\left[ \frac{|\langle u, v \rangle - \langle h(u), h(v) \rangle|}{|\langle u, v \rangle|} \geq \epsilon/c \right] \leq 2 \exp(-(\epsilon^2 - \epsilon^3)k/4)$$

*Proof.* From the norm preservation in Lemma D.2, we know that with probability at least $1 - 2 \exp(-(\epsilon^2 - \epsilon^3)k/4)$,

$$(1 - \epsilon)\|u + v\|^2 \leq \|h(u + v)\|^2 \leq (1 + \epsilon)\|u + v\|^2$$
$$(1 - \epsilon)\|u - v\|^2 \leq \|h(u - v)\|^2 \leq (1 + \epsilon)\|u - v\|^2$$

So, we can write

$$\begin{aligned}
4h(u) \cdot h(v) &= \|h(u + v)\|^2 - \|h(u - v)\|^2 \\
&\geq (1 - \epsilon)\|u + v\|^2 - (1 + \epsilon)\|u - v\|^2 \\
&= 4u \cdot v - 2\epsilon(\|u\|^2 + \|v\|^2) \\
&\geq 4u \cdot v - 4B^2\epsilon
\end{aligned}$$

This implies $\langle u, v \rangle - \langle h(u), h(v) \rangle \leq B^2\epsilon$. So,

$$\frac{\langle u, v \rangle - \langle h(u), h(v) \rangle}{|\langle u, v \rangle|} \leq \frac{B^2\epsilon}{cB^2} = \epsilon/c$$

The other side of the double-sided bound can be derived analogously.   $\square$

We can now look at LoRA for a simple fully connected layer.

| coordinate-wise scale | $M = U^i$ | $M = W^j$ | $M = V$ | $M = W'$ |
|---|---|---|---|---|
| $M_{-1}$ | $\Theta(1)$ | $\Theta(1/\sqrt{n})$ | $\Theta(1/n)$ | - |
| $\Delta M_0$ | $\Theta(1)$ | $\Theta(1/n)$ | $\Theta(1/n)$ | $\Theta(1)$ |
| $\Delta M_t$ | $\Theta(1/\sqrt{n})$ | $\Theta(1/n\sqrt{n})$ | - | - |
| $\eta_M$ for SGD | $\Theta(1)$ | $\Theta(1/n)$ | - | - |
| $\eta_M$ for SignGD/Adam | $\Theta(1/\sqrt{n})$ | $\Theta(1/n\sqrt{n})$ | - | - |

Table 11: Scales for prompt-based fine-tuning with projection layer w.r.t. $n$

**Lemma D.4** (LoRA SGD Kernel). *Let $h = Wx + BAx$ as defined in the paper, where $x \in \mathbb{R}^n$, $W \in \mathbb{R}^{m \times n}$, $B \in \mathbb{R}^{m \times k}$, and $A \in \mathbb{R}^{k \times n}$ with $k \ll n$. $B$ is initialized to 0 and $A$ is initialized with i.i.d. mean-0 Gaussian samples. SGD Training with LoRA (i.e., fixing $W$ and allowing $A$ and $B$ to be updated) yields the kernel $\mathcal{K}_{LoRA}^{(SGD)}$, whereas full FT with SGD yields the kernel $\mathcal{K}$:*

$$\mathcal{K}_{LoRA}^{(SGD)} = dH dH^\top \odot (XA^\top A X^\top) \qquad \mathcal{K}^{(SGD)} = dH dH^\top \odot (XX^\top)$$

*where $dH \in \mathbb{R}^{N \times m}$ has $dh(x_i)$ in the ith row and $X \in \mathbb{R}^{N \times d}$ has $x_i$ in the ith row.*

*Proof.* We start by noting the well-known fact that $dW = dh \otimes x$, where $dh$ is the gradient to $h$ and $\otimes$ is the cross product. Thus, $K = dH dH^\top \odot (XX^\top)$. In the LoRA setting, $dA = 0$ and $dB = dh \otimes Ax$. Because we are in the kernel setting, $B = 0$ and thus, $dA = 0$, throughout training. So,

$$\mathcal{K}_{\text{LoRA}}(i,j) = \langle dB(i), dB(j) \rangle = \langle dh(i), dh(j) \rangle \langle Ax_i, Ax_j \rangle$$

where $dB(a)$ denotes the gradient to $B$ when given example $a$. Analogous reasoning yields

$$\mathcal{K}^{(\text{SGD})}(i,j) = \langle dh(i), dh(j) \rangle \langle x_i, x_j \rangle$$

$\square$

**Theorem D.5** ($\mathcal{K}_{\text{LoRA}}^{(SGD)}$ is likely not far from $\mathcal{K}^{(SGD)}$). *Let $\mathcal{K}_{LoRA}^{(SGD)} \in \mathbb{R}^{N \times N}$ and $\mathcal{K}^{(SGD)} \in \mathbb{R}^{N \times N}$ be defined as in Lemma D.4. Additionally, assume that $|\langle x(\xi), x(\xi') \rangle| \geq c \|x(\xi)\| \|x(\xi')\|$ for some $c > 0$ for all pairs $\xi, \xi'$ in the downstream dataset (i.e., no two gradients are not orthogonal). Then, for any $i, j \in [N]$,*

$$\Pr\left[ \frac{|\mathcal{K}_{LoRA}^{(SGD)}(i,j) - \mathcal{K}^{(SGD)}(i,j)|}{|\mathcal{K}^{(SGD)}(i,j)|} \geq \epsilon/c \right] \leq \exp(-(\epsilon^2 - \epsilon^3)k/4)$$

*Proof.* Note that

$$\frac{\mathcal{K}_{\text{LoRA}}^{(\text{SGD})}(i,j) - \mathcal{K}^{(\text{SGD})}(i,j)}{|\mathcal{K}^{(\text{SGD})}(i,j)|} = \frac{\langle dh(i), dh(j) \rangle (\langle Ax_i, Ax_j \rangle - \langle x_i, x_j \rangle)}{\langle dh(i), dh(j) \rangle \langle x_i, x_j \rangle}$$

$$= \frac{\langle Ax_i, Ax_j \rangle - \langle x_i, x_j \rangle}{\langle x_i, x_j \rangle}$$

The rest of the proof follows from Lemma D.3. $\square$

The statement for IntrinsicDimension FT can be derived by applying Johnson-Lindenstrauss directly to the gradient vectors.

