# OpenReview forum: "A Kernel-Based View of Language Model Fine-Tuning"
_ICLR.cc/2023/Conference — Submitted to ICLR 2023_

### Official Review · Reviewer_NNgP · 2022-10-25

**Confidence:** 2
**Correctness:** 3
**Technical Novelty And Significance:** 2
**Empirical Novelty And Significance:** 3
**Recommendation:** 6

**Clarity, Quality, Novelty And Reproducibility:**

The paper is not well organized. Claim many things but the results are spread out here and there. It's very hard to understand the contribution before linking them. In terms of novelty, I think the angle to approach the underlying NLP phenomena is interesting but no real technical difficulties or insightful theoretical results given.


**Strength And Weaknesses:**

Strength:

- An interesting observation and continuation of previous work.
- Provide studies over various related topics with mixed empirical and theoretical analysis.

Weakness:

Main contribution seems to just be combining two major previous results and somehow it's not providing enough insight.

**Summary Of The Paper:**

This paper studies behavior of NTK on fine-tuning MLM models. Supposedly, pre-trained model is no longer iid random in parameters but still it can achieve good performance on CV tasks per previous literature. In this paper, authors extended the studies to NLP models with both regular fine-tuning and prompt-based learning cases. Authors also provide a modified eNTK version when using ADAM but not SGD. Last, authors tried to propose some hypothesis on why such model on eNTK behaves so.

**Summary Of The Review:**

Overall, I like the angles and the topic of this paper. But somehow I think it's a bit disappointing after reading it as it's not providing much insight. Rather, it's more of a mixed of empirical evidence and some not that useful theoretical results. I do believe we should encourage this type of work so overall I am more positive, but there are some confusions and I hope authors could explain it.

Overall, I think the gist of the paper is a bit unclear. Mentioned too many points but I can't find the direct link between the claim and analysis. You listed many contributions in bullet points so I guess it's easier to go from there.

- Does the eNTK method directly work for NLP tasks? In Section 5, we show that eNTK does not
work well for standard FT but it achieves performance comparable to prompt-based FT (Schick
& Schutze, 2021; Gao et al., 2021) on a majority of downstream tasks that we evaluate. ¨

- So I think this is just an empirical analysis. Somehow just an extension of domain from wei el. al.
Do you agree with this? If so, can you highlight this is more of an empirical results. in addition,  can
you also greatly re-organize your paper to make contribution of empirical results and theoretical analysis clear?

• The NTK theory was developed for gradient descent, whereas LM fine-tuning often uses
Adam (Kingma & Ba, 2014) as a standard practice (Devlin et al., 2019). Can the above method
be extended to give insight into the effect of different optimization algorithms? In Section 4,
we derive a new asymmetric “kernel” formula (Definition 4.2) that describes the dynamics of
short-term training with Adam and shows that the corresponding eNTK achieves comparable
performance as FT using Adam (Table 1).

- Here you use yet another approximation of Adam, so I don't see the purpose of it. Why does that
matter to analyze yet another version of eNTK  just due to a different gradient method?
In particular, Adam results in Table 1 doesn't really outperform SGD version eNTK musch. Especially,
for TREC ADAM result is much worse than eNTK one. What's the explanation?

• Is the eNTK regression method simply an alternate model that also happens to have good
performance on the classification task, or is it actually a near-equivalent description of how
parameters of the original model evolve during fine-tuning? Section 5.3 shows that for tasks that
the eNTK can solve, the fine-tuning of the pre-trained model does exhibit behavior consistent
with the dynamics of training kernel classifiers. See Definition 3.1.

- I think this could be a real contribution. But somehow I feel not consistent as fine-tuning + eNTK
seems not to work but here you showed its empirical performance justified the assumptions/desired
behavior of NTK in 3.1 ? It's kind of contradicting.

• If the eNTK can solve NLP tasks, then does the eNTK give insight into phenomena such as why
prompting can improve performance and how parameter-efficient fine-tuning methods affect
optimization? Figure 1 demonstrates that adding a prompt is necessary for the eNTK to solve the
task, and in Section 6, we apply the kernel lens to provide a possible explanation for the empirical
success of subspace-based fine-tuning methods (Hu et al., 2021; Aghajanyan et al., 2021).

- Sorry I don't quite get why prompting can improve performance is related to subspace-based fine-tuning methods?
I think the connection is not very obvious to me. I was feeling strange when I read your section 6. I am not sure
why you want to analyze it. And I felt the connection to other part of paper is too weak such that it's another paper.

We conclude by proposing a rigorous mechanism through which fine-tuning of complex architectures
(e.g., Transformers (Vaswani et al., 2017)) with prompts can exhibit kernel behavior. This is done
in context of networks whose width goes to infinity, but unlike standard infinite-width NTK theory
it allows a non-random initialization that is the result of pretraining. This result uses the Tensor
Programs framework (Yang, 2019; 2020a;b; Yang & Littwin, 2021; Yang & Hu, 2021).

- I actually feel this part of analysis is the most pertinent part regarding your motivations. Why don't you put it in the front? But I am not sure the correctness and framework as I didn't know Tensor Programs before.

---

> ### Author Response · Authors · 2022-11-15
> **Response to Reviewer NNgP [1/2]**
>
> We thank the reviewer for the careful reading and thoughtful feedback. Please see our general response above for a clarification on empirical and theoretical contributions, and our response to other questions and comments below. We divided the response into two posts to satisfy the character limit.
>
> **Is the eNTK performance just an empirical analysis?**
>
> Yes, the performance of the eNTK and the verification of the Linearization and Fixed Features properties of kernel behavior (Definition 3.1) in Section 5 is an extensive empirical analysis. Experiments differ from  [1] in the following ways: (1) we compute the eNTK on downstream NLP tasks; (2) we compute kernel analogs for Adam (Section 4); (3) we compare the kernel analogs of prompt-based and standard FT; (4) we more directly analyze if fine-tuning exhibits kernel behavior when the eNTK can solve the task (Section 5.3). The goal is to measure when and to what extent language model fine-tuning methods exhibit kernel behavior.
>
> **Why is it important to analyze another version of eNTK corresponding to a different gradient method?**
>
> The kernel analog (Definition 3.2) depends on the optimization method used. The standard NTK formula (Definition 3.3, [2]) corresponds to training with SGD or gradient descent. Previous work in [3, 4, 5] has revealed that using the Adam optimizer is crucial for training transformers. Therefore, we derive the correct kernel analog, i.e. Asymmetric SignGD kernel in Definition 4.2 and Theorem 4.3, for training with the Adam optimizer. We note that developing a kernel analog for Adam is a theoretical contribution of this paper that can apply beyond the scope of studying fine-tuning. A unique feature of the SignGD kernel is its invariance to the relative scales of the gradients, which may be a desirable property warranting further theoretical and empirical exploration.
>
> **Table 1 shows that SGD and Adam perform comparably for prompt-based fine-tuning, so what is the reasoning to study the kernel analog of Adam?**
>
> Our discovery that SGD and Adam often perform comparably in the prompt-based fine-tuning is new and contradictory to existing belief that Adam is required for fine-tuning on transformers. Although SGD and Adam achieve comparable performance, one algorithm may exhibit kernel behavior while the other does not, so we check the eNTK performance for all three kernels and compare them to the analogous fine-tuning procedure. Comparing the performance of the different kernels to each other does not have a well-understood implication, because each kernel analog is meant to mirror its respective fine-tuning procedure.
>
> **What is the logic around this point from the introduction? “Is the eNTK regression method simply an alternate model that also happens to have good performance on the classification task, or is it actually a near-equivalent description of how parameters of the original model evolve during fine-tuning? Section 5.3 shows that for tasks that the eNTK can solve, the fine-tuning of the pre-trained model does exhibit behavior consistent with the dynamics of training kernel classifiers. See Definition 3.1.”**
>
> We state that the experiments directly verify that training can often (but not always) be described with kernel dynamics. In particular, the definition of kernel behavior in Definition 3.1 highlights two properties (i.e, Linearization and Fixed Features) that must hold, and the definition is well-established from previous works on the NTK. Section 5.3 presents experiments that directly check these two properties, and we conclude from the results that fine-tuning on many tasks exhibits kernel behavior. We do not use the experiments in Section 5.3 to define what kernel behavior is – the properties of kernel-based dynamics are well defined and studied in many prior NTK works.
>
> *References*
>
> [1] Wei, Alexander, Wei Hu, and Jacob Steinhardt. "More Than a Toy: Random Matrix Models Predict How Real-World Neural Representations Generalize." ICML 2022.
> https://proceedings.mlr.press/v162/wei22a/wei22a.pdf
>
> [2] Jacot, Arthur, Franck Gabriel, and Clément Hongler. "Neural tangent kernel: Convergence and generalization in neural networks."  NeurIPS 2022.
> https://proceedings.neurips.cc/paper/2018/file/5a4be1fa34e62bb8a6ec6b91d2462f5a-Paper.pdf
>
> [3] Zhang, Jingzhao, Sai Praneeth Karimireddy, Andreas Veit, Seungyeon Kim, Sashank Reddi, Sanjiv Kumar, and Suvrit Sra. "Why are adaptive methods good for attention models?." NeurIPS 2020.
> https://proceedings.neurips.cc/paper/2020/file/b05b57f6add810d3b7490866d74c0053-Paper.pdf
>
> [4] Liu, Liyuan, Xiaodong Liu, Jianfeng Gao, Weizhu Chen, and Jiawei Han. "Understanding the difficulty of training transformers." EMNLP 2020
> https://aclanthology.org/2020.emnlp-main.463.pdf
>
> [5] Li, Zhiyuan, Srinadh Bhojanapalli, Manzil Zaheer, Sashank Reddi, and Sanjiv Kumar. "Robust training of neural networks using scale invariant architectures." ICML 2022.
> https://proceedings.mlr.press/v162/li22b/li22b.pdf

---

> ### Author Response · Authors · 2022-11-15
> **Response to Reviewer NNgP [2/2]**
>
> **How does the finding that prompting can improve performance relate to subspace-based fine-tuning methods?**
>
> We believe our writing has caused some confusion on this point. We mean to say that only prompt-based fine-tuning exhibits kernel behavior. Additionally, when training exhibits kernel behavior, it is straightforward to understand how subspace-based fine-tuning methods impact optimization. Therefore, we can rigorously characterize subspace-based fine-tuning methods only when using a prompt. We revised the writing in the overview of the introduction and Section 6 to clarify this point.
>
> **Why did you analyze subspace-based fine-tuning methods? The connection to the other part of the paper is too weak.**
>
> We analyze subspace-based fine-tuning methods as a demonstration of how a kernel-based view of fine-tuning can aid empiricists and theoreticians in the design and analysis of new efficient fine-tuning methods. Parameter-efficient fine-tuning methods have become an active area of research in NLP [6]. These methods reduce the memory overhead of standard fine-tuning methods by requiring only a small number of task-specific parameters to be trained and stored for inference. Despite their empirical success and desirable properties, these methods are poorly understood. However, the kernel view admits a clear and simple interpretation of how these methods modify the optimization trajectory during fine-tuning. We present only the most straightforward application of the kernel view but note that many variants and fine-tuning “tricks” modify the kernel in mathematically meaningful ways. We leave it to future work to account for the success of such methods when fine-tuning does not exhibit kernel behavior.
>
> *References*
>
> [6] He, Junxian, Chunting Zhou, Xuezhe Ma, Taylor Berg-Kirkpatrick, and Graham Neubig. "Towards a unified view of parameter-efficient transfer learning." ICLR 2022. https://openreview.net/forum?id=0RDcd5Axok

---

### Official Review · Reviewer_rpG6 · 2022-10-28

**Confidence:** 4
**Correctness:** 3
**Technical Novelty And Significance:** 3
**Empirical Novelty And Significance:** 2
**Recommendation:** 5

**Clarity, Quality, Novelty And Reproducibility:**

Clarity & Quality: The paper is overall clear, I'd be happy to get clarified on Section 4 & 7 a bit more. In terms of writing, I do think there is a better way to re-organize and better separate the assumption & theory with empirical results. Currently, empirical evidence and theory are interleaving and a bit hard to get the full picture.

Novelty & Reproducibility: Overall relatively novel and should be easy to reproduce.

**Strength And Weaknesses:**

Strength:
The paper tries to explain language model fine-tuning using NTK and cleverly uses the fact that fine-tuning takes few samples (therefore one can compute eNTK in a feasible way).

Weaknesses:
Overall, the paper is trying to bridge theory and practice. But there are a few concerns:
1. While some of the empirical results are good, some observations are quite mixed. For example, RTE linearization does not perform well for k=512 while eNTK still performs ok -- this seems confusing to me.
2. Section 4 is unclear to me, the paper mentioned "we model Adam updates with SignGD", why can you replace Adam with SignGD and is the derived kernel exactly following Adam or is just an approximation of Adam (if later is the case, why not just approximate Adam with SGD)?
3. The theory section (Section 7): I don't see how it is related to prompt-based fine-tuning but not standard fine-tuning. The paper mentioned "The theory focuses on the prompt-based setting since we did not find empirical evidence of kernel behavior in the standard setting" (I hope theory can advance empirical work but maybe now the paradigm is shifted), this is fine, but which assumption in Theorem 7.3 is specific to prompt such that the final conclusion is "prompt-based FT of f will exhibit kernel behavior"?
4. The bigger question is whether NTK is THE theory or just A theory. Obviously a lot of practices in LM training or FT do not match with NTK behavior (e.g., learning rate warmup), therefore assuming NTK is THE theory and develop reasoning that tries to explain FT is a bit insufficient, so I think this is a weakness of the paper.

**Summary Of The Paper:**

The paper suggests that the success of language model fine-tuning (or with prompt) can be explained by neural tangent kernel (NTK), a tool that describes optimization behavior of infinitely wide neural networks. To bridge NTK with LM fine-tuning, the paper tries to justify that using NTK after pre-training phase (instead of initializing from lazy regime) is reasonable, and also derived empirical NTK for adam optimizer. The paper found empirical evidence that NTK behaves similarly to the original optimization methods, and also provided a theorem that shows prompt-based fine-tuning can exhibit kernel behavior

**Summary Of The Review:**

My feeling is a bit mixed for this paper. My current rating would be marginally below the threshold given that there are several issues worth clarifying. Regarding Weakness #4, I think different people may have different feeling on how important this is. I am relatively close to the median of that spectrum, but I can definitely see researchers having strong opinions from both sides.

---

> ### Author Response · Authors · 2022-11-15
> **Response to Reviewer rpG6**
>
> We would like to thank the reviewer for the careful reading and thoughtful feedback. Please see our general response above, and our response to other questions and comments below. We have revised the overview at the end of the introduction and clarified our logic throughout the paper. Please find our highlighted edits in the revision.
>
> **“Is NTK the description of FT or just one possible mechanism?”**
> Please see our general response.
>
> **“...some observations are quite mixed. For example, RTE linearization does not perform well for k=512 while eNTK still performs ok -- this seems confusing to me.”**
>
> In general, when the eNTK can solve a task but does not exhibit kernel behavior (i.e., either the linearization or the fixed features property appears not to hold), this implies that the kernel is a good alternative model for solving the task, but fine-tuning does not obey kernel-based dynamics. Such a result may be useful if one wants to replace fine-tuning with kernel regression, but it does not allow one to apply kernel-based theoretical results (e.g., generalization bounds in [0]) to the fine-tuning procedure.
>
> We would like to point out that Table 1 has results for k = 16 and k = 64 and Figure 2 has results for k = 16 and **k = 512**. For k = 16, the results are quite consistent for RTE. However, when k increases, the eNTK performs worse and the linearization property does not seem to hold. This trend is expected, because fine-tuning is less likely to exhibit kernel behavior when more gradient steps are taken. We are sorry about the confusion and will make the presentation more consistent in the revision (although we believe the results are consistent).
>
> **Why can you replace Adam with SignGD? Is the derived kernel exactly following Adam or just an approximation of Adam? If the latter is the case, why not approximate Adam with SGD?**
>
> Past works have shown SignGD to be the correct approximation for early-stage Adam training. Propositions 1 and 3 in [1] showed that full-batch Adam behaves like SignGD when the learning rate is small and relatively few gradient steps are taken. In this case, the second order moment estimate and the momentum terms will compute moving averages in a small neighborhood, so the normalized Adam updates can be rewritten as SignGD. [2] reached an analogous conclusion for stochastic mini-batch Adam, which is the common choice for language model fine-tuning.
>
> SGD is not a good approximation for Adam because Adam performs normalization before applying updates, and SGD is sensitive to the scale of the gradients. The derived asymmetric kernel (Definition 4.2 and Theorem 4.3) corresponds to training with an approximation of Adam (i.e., SignGD), but it has been shown theoretically and empirically in the above papers that this approximation is much better than SGD and indeed quite descriptive of the early-stage dynamics of Adam. We have updated the paper in Section 4 to include an expanded discussion of the approximation and the significance of the resulting kernel.
>
> **What assumptions in the theoretical analysis (Section 7) apply only to prompt-based fine-tuning and not to standard fine-tuning?**
>
> Using a prompt for the downstream task turns it into a fill-in-the-blank problem, which allows it to be viewed as a subcase of the pre-training task.  Our theoretical analysis assumes that an infinitely wide net solves the downstream task **perfectly with no fine-tuning**. This assumption seems too strong when the downstream task does not involve a prompt, because then it would be solved by adding a head that is randomly initialized, and the head would also have infinitely many parameters. We have substantially updated the writing in Section 7.
>
> **Using learning rate warmup contradicts kernel behavior.**
>
> Although a few papers report using learning rate warmup during fine-tuning, there is not a clear empirical or theoretical understanding of whether it is necessary. In our paper, we do not use learning rate warmup and achieve competitive scores. Also, we do **not** claim that **all** fine-tuning methods display kernel behavior.
>
> *References*
>
> [0] Arora, Sanjeev, Simon Du, Wei Hu, Zhiyuan Li, and Ruosong Wang. Fine-grained analysis of optimization and generalization for overparameterized two-layer neural networks, ICML 2019. https://proceedings.mlr.press/v97/arora19a.html
>
> [1] Ma, Chao, Lei Wu, and E. Weinan. "A Qualitative Study of the Dynamic Behavior for Adaptive Gradient Algorithms." In Mathematical and Scientific Machine Learning, pp. 671-692. PMLR, 2022.
> https://proceedings.mlr.press/v145/ma22a/ma22a.pdf
>
> [2] Malladi, Sadhika, Kaifeng Lyu, Abhishek Panigrahi, and Sanjeev Arora. On the SDEs and Scaling Rules for Adaptive Gradient Algorithms, NeurIPS 2022.
> https://openreview.net/forum?id=F2mhzjHkQP

---

### Official Review · Reviewer_Ftva · 2022-11-03

**Confidence:** 2
**Correctness:** 2
**Technical Novelty And Significance:** 2
**Empirical Novelty And Significance:** 3
**Recommendation:** 6

**Clarity, Quality, Novelty And Reproducibility:**

I found the paper unclear and hard to follow, in a way that is hard to appropriately appreciate the results. In particular I found the following points unclear:
- Framing: The paper aims to further an empirical and theoretical understanding of pre-training and fine-tuning paradigm for NLP tasks, but this goal is too broad. As a result, the paper highlight too many points, and does not focus on none of them, especially with the the page limit. Creating a coherent story by refining a single take-away message from all of the described results, will help readers to follow the paper.
- Mathematical motivation: The mathematical definitions appear without a proper motivation. Although the paper claim that some of the definitions are analogues to definitions in other papers, I believe the paper should be self-contained, and provide the intuition in high level.
- Self contains claims: All mathematical claims should be self contained within the body of the paper, where the appendix can be used for proof, and for a formal version of the claims. Most claims of the paper are indeed presented in a self-contained manner, except Theorem 7.3, in which the conditions themselves appear in the appendix, without any informal phrasing of them.
- references: it is very helpful that the paper have hyperlink to the arXiv, but it shouldn't replace mentioning the peer-review venue in which the papers appeared in. In the current format it is hard to validate that the paper is based on peered-review works.
- Acronyms: please define all acronyms, for examples SGD.



**Strength And Weaknesses:**

Strength:
The paper tackle one of the most important questions in machine learning: understanding the reason behind the the performance of fine-tuned models, and in particular, when it should works.

Weakness
I found it hard to follow the paper's arguments. see section below for more details.

**Summary Of The Paper:**

The paper studies the underlying reason behind why fine tuning pre-trained language models works well. The paper shows that in some cases, those fine tuned models can be described using neural tangent kernel. In those cases, the kernel view provide explanation of the fine-tuning success.


**Summary Of The Review:**

The paper presents several theoretical and experimental results, that I don't feel creates a coherent bottom line. In addition, It is hard to evaluate the quality of those results, without reading the appendix (which I didn't read). Thus, I cannot recommend accepting the paper.

---

> ### Author Response · Authors · 2022-11-15
> **Response to Reviewer Ftva**
>
> We thank the reviewer for careful reading and specific comments on the writing. We updated the references to include the peer-reviewed venue where they were published, added more motivation for the mathematical definitions and the significance of the theorems that we show, and defined any acronyms we use. As suggested, we have revised Section 5 and the introduction to include a concise overview of the key kernel-related takeaways from the empirical analysis: (1) only prompt-based FT exhibits kernel behavior, (2) the eNTK fails to solve multi-class tasks, and (3) entailment tasks demonstrate counterintuitive optimization properties. We have also majorly updated Section 7 to clarify the logic and ultimate goal of our theoretical analysis. Our updates can be found highlighted in the revision.
>
> **The conditions for Theorem 7.3 appear in the appendix without any informal phrasing in the main paper.**
>
> We revised Section 7 of our paper to include an informal discussion of the assumptions in the main text. The network must be (1) stable: the scale of its output must not grow with the width, so that the infinite-width limit is meaningful, and (2) non-trivial: the function must be able to update its value so that learning is possible during fine-tuning. We also require that the nonlinearities in the network are pseudo-Lipschitz and that the SignGD updates behave relatively well. These assumptions are standard for analyses that proceed through Tensor Programs, and the restrictions on SignGD are specific to our work but loose enough to accommodate standard pre-training and fine-tuning schemes.

---

> > ### Comment · Reviewer_Ftva · 2022-11-30
> > **The paper looks much better after revision**
> >
> > I find the revised version much clearer and fluent. The concise overview of Section 5, together with the motivation of Section 7, are very helpful to get a coherent bottom line that is supported both by theory and experiments that prompt-based FT (often/ should) exhibits kernel behavior. After understanding this point, Section 6 become interesting to me, as it demonstrates that indeed kernel behavior is a useful tool towards theoretical understanding.
> >
> > I still find the following weaknesses, which are milder than my original expression of the paper:
> > - It is difficult to get an intuitive search of which tasks are Natural Task in the Infinite-Width Limit. Why the paper does a fair job in the description, the mathematical formula does not capture an easy-to-see concept, which is an inherent limitation of the approach.
> > - Results for standard SGD FT do not tell a coherent story. It is great that the paper acknowledges this limitation, but it is still a limitation of the approach.
> >
> > Thus, I am changing my overall evaluation into a weak accept.

---

> > > ### Author Response · Authors · 2022-12-02
> > > **Thank you!**
> > >
> > > Thank you for your response and for raising the score to account for the clearer revision. Below, we address the remaining concerns.
> > >
> > > **It is difficult to get an intuitive search of which tasks are Natural Task in the Infinite-Width Limit.**
> > > We believe this is due to the notation in the definition, which we did not revise yet to avoid confusion during the discussion period. One feature we will highlight in the revision is that the *combination* of a task and a usage method of the pre-trained model (e.g., using a prompt or initializing new parameters) is natural in the infinite-width limit. This will clarify the role of the prompt in the subsequent theoretical analysis as well.
> > >
> > > **Results for standard SGD FT do not tell a coherent story.**
> > > Our experiments in Figure 1 show that standard FT cannot be explained by kernel, and our theory in Section 7 shows how the difference between prompt-based and standard FT can be formally characterized. In particular, our mechanism for fine-tuning assumes that an infinitely wide net solves the downstream task perfectly with no adaptation. This assumption is impossible with standard FT because the head is randomly initialized, thus making standard FT unlikely to exhibit kernel behavior.

---

### Author Response · Authors · 2022-11-15
**General response to all reviewers**

We thank reviewers for their time and thoughtful reviews. We have responded to questions from each reviewer, revised the paper accordingly, and highlighted major changes in the text. Here, we would like to highlight the following comments.

**Is NTK the description of FT or just one possible mechanism? Assuming that the NTK is the correct theoretical description for fine-tuning may result in flawed reasoning.**

We agree. We do **not** claim that **all** fine-tuning methods display kernel behavior; instead, we conduct an extensive set of experiments with standard settings and diverse tasks to find that fine-tuning exhibits kernel behavior in many cases. Our goal is to highlight that fine-tuning is **often** performing a mathematically simple operation on the pre-trained network.

**The writing does not provide a clear take-away message, especially because empirical and theoretical results are interleaved and the mathematical definitions appear without motivation.**

We thank the reviewers for feedback on the writing and organization of the paper. We have updated the overview section of our introduction to highlight our claims clearly and summarize them below.
- We formally extend the standard NTK theory developed for gradient descent to characterize kernel-based dynamics when training with Adam.
- We perform an extensive empirical analysis on 12 diverse NLP tasks and find that fine-tuning can often be described by kernel-based dynamics.
- We straightforwardly apply the kernel view of FT dynamics to formally analyze the success of empirically popular fine-tuning methods that update in a low-rank subspace of model parameters (e.g., LoRA).
- We formally extend infinite-width analysis to account for a pre-trained initialization and characterize conditions under which fine-tuning can exhibit kernel behavior.

For the camera-ready version of the paper, we will put the theoretical analysis (current Section 7) before the subspace-based fine-tuning analysis (current Section 6). To make room for a clearer exposition, we removed the proof sketch in Section 7 and moved the table in the theory section to the appendix.

---

### Author Response · Authors · 2022-11-23
**Requesting further discussion**

Dear reviewers,

We have revised our manuscript based on your valuable and constructive reviews. We would appreciate it if you could check our responses and re-evaluate our paper. We are looking forward to discussing any further questions or suggestions!

Again, thank you for your time and effort in reviewing our paper!

---

### Decision · Program_Chairs · 2023-01-20

**Decision:**

Reject

**Justification For Why Not Higher Score:**

Agreement among all 3 reviewers that the paper is not written well enough and the results aren't conclusive enough. For this type of paper, a clear message is crucial for it to have an impact.

**Justification For Why Not Lower Score:**

n/a

**Metareview: Summary, Strengths And Weaknesses:**

The paper analyzes the process of fine tuning of language models via an analysis using Neural Tangent Kernels. The goal of better understanding the process of fine-tuning is well motivated and the reviews agree that there is value in a result deepening the communities understanding of the process. In addition to being a well motivated problem, the paper has healthy mix of theoretical and empirical analysis.
The paper exhibits two major weaknesses, both raised in all of the reviews. The first regards the clarity of the paper: The reviewers had a hard time understanding the main message of the paper, and the details of the theoretical claims and proofs. The authors improved the writing during the rebuttal stage, but this is acceptable when there are a handful of isolated issues, and this doesn't seem to be the case (for example, in the response to the rebuttal, reviewer Ftva still mention the incoherence of the story as weakness).
A second issue that I find more problematic regards the impact of the paper. Although the reviews do not directly mention it, they all mention that the conclusions aren't coherent, that observations are mixed, or the paper isn't providing enough insight. This may be an issue with the writing clarity, but given that all reviews make these claims in some way, there is a risk of the results not being actionable / insightful enough, making impact a weakness of the paper.
To conclude, the paper in its current state doesn't seem to meet the bar for ICLR.